# Tracing Hyperparameter Dependencies for Model Parsing via Learnable Graph Pooling Network

**Xiao Guo, Vishal Asnani, Sijia Liu, Xiaoming Liu**
Michigan State University
{guoxia11, asnanivi, liusiji5, liuxm}@msu.edu

## Abstract

*Model Parsing* defines the task of predicting hyperparameters of the generative model (GM), given a GM-generated image as the input. Since a diverse set of hyperparameters is jointly employed by the generative model, and dependencies often exist among them, it is crucial to learn these hyperparameter dependencies for improving the model parsing performance. To explore such important dependencies, we propose a novel model parsing method called Learnable Graph Pooling Network (LGPN), in which we formulate model parsing as a graph node classification problem, using graph nodes and edges to represent hyperparameters and their dependencies, respectively. Furthermore, LGPN incorporates a learnable pooling-unpooling mechanism tailored to model parsing, which adaptively learns hyperparameter dependencies of GMs used to generate the input image. Also, we introduce a Generation Trace Capturing Network (GTC) that can efficiently identify generation traces of input images, enhancing the understanding of generated images' provenances. Empirically, we achieve state-of-the-art performance in model parsing and its extended applications, showing the superiority of the proposed LGPN. The source code is available at link.

## 1 Introduction

Generative Models (GMs) [22, 73, 11, 44, 33, 60, 32, 34, 50], *e.g.*, Generative Adversarial Networks (GANs), Variational Autoencoder (VAEs), and Diffusion Models (DMs), have gained significant attention, offering remarkable capabilities in generating visually compelling images. However, the proliferation of such Artificial Intelligence Generated Content (AIGC) can inadvertently propagate inaccurate or biased information. To mitigate such negative impact, various image forensics [52] methods have been proposed [62, 54, 20, 6, 24, 64, 31, 76, 56, 71, 1]. Alongside these defensive efforts, the recent work [2] defines a novel research topic called "*model parsing*", which predicts 37 GM hyperparameters using the generated image as the input, as detailed in Fig. 1a.

Model parsing requires analyzing GM hyperparameters and gaining insights into origins of generated images, which facilitate defenders to develop effective countermeasures. For example, one can reasonably determine if there exists coordinated attacks [2] — two images are generated from the same GM that is *unseen* during the training, using predicted hyperparameters from the model parsing algorithm. In light of this, the previous method [2] introduces a clustering-based approach that achieves effective model parsing performance. However, this approach neglects the learning of hyperparameter dependencies. For instance, an inherent dependency exists between GM's layer number and parameter number, as GM's layer number is positively proportional to its parameter number. A similar proportional relationship exists between the number of convolutional layers and convolutional filters. In contrast, the use of the L1 loss is negatively correlated with the use of the L2 loss, as GMs typically do not employ both losses as objective functions simultaneously. We believe the neglect of such dependencies might cause a suboptimized performance. Therefore, in

38th Conference on Neural Information Processing Systems (NeurIPS 2024).

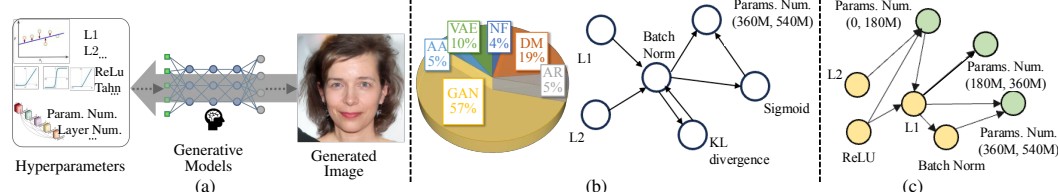

Figure 1: (a) Hyperparameters define a GM that generates images. *Model parsing* [2] refers to the task of predicting hyperparameters given the generated image. (b) We study the co-occurrence pattern among different hyperparameters in various GMs from the RED140 dataset whose composition is shown as the pie chart[1], and subsequently construct a directed graph to capture dependencies among these hyperparameters. (c) We define the discrete-value graph node (▨) (*e.g.*, L1 and `Batch Norm`) for each discrete hyperparameter. For each continuous hyperparameter (▨), we partition its range into $n$ distinct intervals, and each interval is then represented by a graph node: `Parameter Number` has three corresponding continuous-value graph nodes. Representations on these graph nodes are used to predict hyperparameters.

this work, we propose to use graph nodes and edges to explicitly represent hyperparameters and their dependencies, respectively, and then propose a model parsing algorithm that utilizes the effectiveness of Graph Convolution Network (GCN) [35, 61, 18, 51] to capture dependencies among graph nodes.

Specifically, we first use training samples in the RED dataset [2] to construct a directed graph (Fig. 1b). The directed graph, based on the label co-occurrence pattern, illustrates the fundamental correlation between different categories and helps prior GCN-based methods achieve remarkable performances in various applications [10, 7, 68, 46, 16, 58]. In this work, this directed graph is tailored to the model parsing — we define discrete-value and continuous-value graph nodes to represent hyperparameters, shown in Fig. 1c. Then, we use this graph to formulate model parsing as a graph node classification problem, in which the discrete-value graph node feature decides if a given hyperparameter is used in the given GM, and the continuous-value node feature decides which range the hyperparameter resides. This formulation helps obtain effective representations of hyperparameters and dependencies among them for the model parsing task, detailed in Sec. 3.1.

To this end, we propose a novel model parsing framework called Learnable Graph Pooling Network (LGPN), which contains a Generation Trace Capturing Network (GTC) and a GCN refinement block (Fig. 2). Our GTC differs from neural network backbones used in State-of-The-Art (SoTA) forgery detection methods [62, 74, 63, 4, 63], using down-sampling operations (*e.g.*, pooling) gradually reduce the learned feature map resolution during the forward propagation. This down-sampling can cause the loss of already subtle generation traces left by GMs. Instead, our GTC leverages a high-resolution representation that largely preserves generation traces throughout the forward propagation (Sec. 3.2.1). Therefore, the learned image representation from the GTC deduces crucial information (*i.e.*, generation trace) of used GMs and benefits model parsing. Subsequently, this representation is transformed into a set of graph node features, along with the pre-defined directed graph, which are fed to the GCN refinement block. The GCN refinement block contains trainable pooling layers that progressively convert the correlation graph into a series of coarsened graphs by merging original graph nodes into supernodes. Then, the graph convolution is conducted to aggregate node features at all levels of graphs, and trainable unpooling layers are employed to restore supernodes to their corresponding children nodes. This learnable pooling-unpooling mechanism helps LGPN generalize to parsing hyperparameters in unseen GMs and improves the GCN representation learning. In summary, our contributions are:

◇ We innovatively formulate *model parsing* as a graph node classification problem, using a directed graph to help capture hyperparameter dependencies for better model parsing performance.

◇ A learnable pooling-unpooling mechanism is introduced with GCN to enhance representation learning in model parsing and its generalization ability.

◇ We propose a Generation Trace Capturing Network (GTC) that utilizes high-resolution representations to capture generation traces, facilitating a deeper understanding of the image's provenance.

---

[1]We adhere to naming conventions from the previous work [2], as the pie chart of Fig. 1b, where AA, AR, and NF represent Adversarial Attack models, Auto-Regressive models, and Normalizing Flow models, respectively.

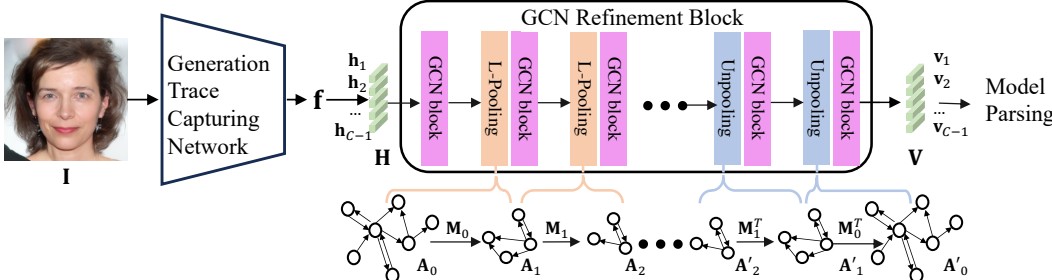

Figure 2: **Learnable Graph Pooling Network**. Given an input image $\mathbf{I}$, the proposed LGPN first uses the Generation Trace Capturing Network (Fig. 3) to extract the representation $\mathbf{f}$. Then, $\mathbf{f}$ is transformed into $\mathbf{H}$, which represents a set of graph node features and is fed into the GCN refinement block. The GCN refinement block stacks GCN layers with paired pooling-unpooling layers (Sec. 3.2.2) and produces the refined feature $\mathbf{V}$ for model parsing. Our method is jointly trained with 3 different objective functions (Sec. 3.3).

◇ Extensive empirical results demonstrate the SoTA performance of the proposed LGPN in model parsing and identifying coordinated attacks. Additionally, the GTC's effectiveness is validated through CNN-generated image detection and image attribution tasks.

## 2 Related Works

**Model Parsing** Previous model parsing works [59, 30, 17, 3, 47] require prior knowledge of machine learning models and their inputs to predict model hyperparameters, and such predictions are primarily limited to architecture-related hyperparameters [17, 47]. In contrast, Asnani *et al.* [2] recently propose a technique to estimate 37 pre-defined hyperparameters covering loss functions and architectures by only leveraging generated images as inputs. Specifically, this work [2] designs FEN-PN that uses a clustering-based approach to estimate the mean and standard deviation of hyperparameters for network architecture and loss function types. Then, FEN-PN uses a fingerprint estimation network that is trained with four constraints to estimate the fingerprint for each image. These fingerprints are employed to predict hyperparameters. However, FEN-PN overlooks dependencies among hyperparameters, which cannot be adequately captured by the estimated mean and standard deviation. In contrast, we propose LGPN, using GCN to model dependencies among different hyperparameters, improving overall model parsing performance.

**GCN-based Method** Graph Convolution Neural Network (GCN) shows effectiveness in encoding dependencies among different graph nodes [35, 61, 18, 26, 28], and in the computer vision community, directed graphs based on label co-occurrence patterns are used with GCN in tasks such as multi-label image recognition [10, 7, 68], semantic segmentation [9, 29, 41, 16], and person ReID and action localization [46, 65, 8, 58]. These works rely on original graph structures, whereas our method uses a pooling algorithm to modify this graph structure, which can benefit GCN's representation learning. Also, our GCN refinement block differs from Graph U-Net [19] in that it reduces the graph size by removing certain nodes. However, we formulate model parsing as a graph node classification task, where each node represents a specific hyperparameter, meaning no nodes should be discarded.

**Learning Image Generation Traces** GMs leave particular traces in their generated images [43, 12], which can be visual artifacts [72, 4] or through evident peaks in the frequency domain [74, 62]. These traces serve as important clues for image forensic tasks such as detection [12, 74, 62], attribution [70, 49] and model parsing [2, 66, 67, 21]. Current methods [62, 74, 63, 4, 63, 57] use backbones like ResNet and XceptionNet that have high generalization abilities, and vision-language foundation models [55, 75, 23] also show effectiveness in detecting unseen forgeries. However, these prior methods often use backbones that rely on downsampling operations, such as convolutions with large strides and pooling, which capture global semantics but discard high-frequency details that contain critical generation traces. In contrast, we propose a Generation Trace Capturing network that mainly operates on a high-resolution representation for capturing such generation traces. Its effectiveness is shown in our experiment by comparing against recent detection methods that use pre-trained CLIP features [48] and the novel representation [63], and competitive image attribution methods.

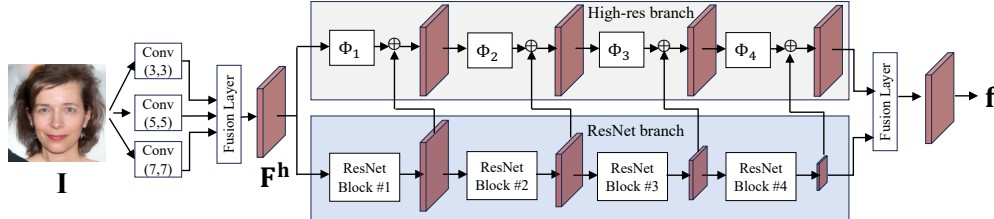

Figure 3: **Generation Trace Capturing Network**. First, convolution layers with different kernel sizes extract feature maps of the input image **I**. A fusion layer concatenates these feature maps and then proceeds the concatenated feature to the ResNet branch and High-res branch.

## 3 Method

In this section, we first revisit some fundamental preliminaries in Sec. 3.1, and then introduce the proposed Learnabled Graph Pooling Network (LGPN) in Sec. 3.2. Lastly, we describe training and inference procedures in Sec. 3.3.

### 3.1 Preliminaries

This section provides the problem statement of *model parsing* and its formulation as a graph node classification task with the Graph Convolutional Network (GCN).

**Revisiting Model Parsing** Given an input image $\mathbf{I} \in \mathbb{R}^{3 \times W \times H}$ generated by a GM (*i.e.*, $\mathcal{G}$), the model parsing algorithm generates three vectors ($\mathbf{y}^d \in \mathbb{R}^{18}$, $\mathbf{y}^c \in \mathbb{R}^9$, and $\mathbf{y}^l \in \mathbb{R}^{10}$) as predictions of $G$ different hyperparameters used in the $\mathcal{G}$. Specifically, defined by the previous work [2], these predictable $G$ hyperparameters include discrete and continuous architecture hyperparameters, as well as loss functions, which are denoted as $\mathbf{y}^d$, $\mathbf{y}^c$, and $\mathbf{y}^l$, respectively. For $\mathbf{y}^d$ and $\mathbf{y}^l$, each element is a binary value representing if the corresponding feature is used or not. Each element of $\mathbf{y}^c$ is the value of certain continuous architecture hyperparameters, such as the layer number and parameter number. As these hyperparameters are in different ranges, we normalize them into [0, 1], same as the previous work [2]. Predicting $\mathbf{y}^d$ and $\mathbf{y}^l$ is a classification task while the regression is used for $\mathbf{y}^c$. Detailed definitions are in Tab. 4, 5, and 6 of the Supplementary. We augment the previous model parsing dataset (*e.g.*, RED116 [2]) with different diffusion models such as DDPM [27], ADM [15] and Stable Diffusions [50], increasing the spectrum of GMs. Also, we add real images on which these GMs are trained into the dataset. In the end, we collect $140$ GMs in total and denote such a collection as the RED140 dataset. Details are in the Supplementary Sec. D.

**Correlation Graph Construction** We design a graph structure of model parsing, where graph nodes and edges represent hyperparameters and their dependencies, respectively. We first use conditional probability $P(L_j|L_i)$ to denote the probability of hyperparameter $L_j$ occurrence when hyperparameter $L_i$ appears. We count the occurrence of such pairs in the RED140 to construct the matrix $\mathbf{G} \in \mathbb{R}^{C \times C}$, where $C$ and $\mathbf{G}_{ij}$ denotes the number of graph nodes and the conditional probability of $P(L_j|L_i)$, respectively. Next, we apply a fixed threshold $\tau$ to remove edges with low correlations in $\mathbf{G}$ and then obtain a directed graph, denoted as $\mathbf{A} \in \mathbb{R}^{C \times C}$, where $\mathbf{A}_{ij}$ indicates if there exists an edge between node $i$ and $j$.

Specifically, as depicted in Fig. 1c, each discrete hyperparameter (*e.g.*, discrete architecture hyperparameters and loss functions) is represented by one graph node of $\mathbf{A}$, denoted as a discrete-value graph node. For continuous hyperparameters, we first divide its value range into $n$ different intervals, and each interval is represented by one graph node of $\mathbf{A}$ denoted as a continuous-value graph node. In other words, $C$ is larger than $G$ since each continuous hyperparameter is represented by $n$ continuous-value graph nodes. Subsequently, we use discrete-value graph nodes to decide if given hyperparameters are present, and the continuous-value node decides which range the hyperparameter resides. Therefore, we denote the constructed graph as $\mathbf{A}$ and apply stacked graph convolution on $\mathbf{A}$ as follows:

$$\mathbf{h}_i^l = \text{ReLU}(\sum_{j=1}^{C} \mathbf{A}_{i,j} \mathbf{W}^l \mathbf{h}_j^{l-1} + \mathbf{b}^l), \tag{1}$$

where $\mathbf{h}_i^l$ represents the $i$-th node feature in graph $\mathbf{A}$. $\mathbf{W}^l$ and $\mathbf{b}^l$ are weight and bias terms.

## 3.2 Learnable Graph Pooling Network

In this section, we detail the Generation Trace Capturing Network (GTC) and GCN refinement block—two major components used in the proposed LGPN, as depicted in Fig. 2.

### 3.2.1 Generation Trace Capturing Network

As shown in Fig. 3, GTC uses one branch (*i.e.*, ResNet branch) to propagate the original image information. Meanwhile, the other branch, denoted as *high-res branch*, harnesses the high-resolution representation that helps detect high-frequency generation artifacts stemming from various GMs. More formally, three separate 2D convolution layers with different kernel sizes (*e.g.*, $3 \times 3$, $5 \times 5$ and $7 \times 7$) are utilized to extract feature maps of $\mathbf{I}$. We concatenate these feature maps and feed them to the fusion layer — the $1 \times 1$ convolution for the channel dimension reduction. Then, we obtain the feature map $\mathbf{F}^h \in \mathbb{R}^{D \times W \times H}$, with the same resolution as $\mathbf{I}$. After that, we proceed $\mathbf{F}^h$ to a dual-branch backbone. Specifically, we upsample intermediate features output from each ResNet block and incorporate them into the high-res branch, as depicted in Fig. 3. The high-res branch also has four different convolution blocks (*e.g.*, $\Phi_b$ with $b \in \{1 \dots 4\}$), which do not employ down-sampling operations, such as the 2D convolution with large strides or pooling layers. Then, intermediate feature maps throughout the high-res branch possess the same resolution as $\mathbf{F}^h$.

The ResNet branch and high-res branch output feature maps are concatenated and then passed through an `AVGPOOL` layer. Then, we obtain the final learned representation, $\mathbf{f} \in \mathbb{R}^D$, that captures generation artifacts of the input image $\mathbf{I}$. Subsuqently, we learn $C$ independent linear layers, *i.e.*, $\Theta = \{\theta_{i=0}^{C-1}\}$ to transform $\mathbf{f}$ into a set of graph node features $\mathbf{H} = \{\mathbf{h}_0, \mathbf{h}_1, ..., \mathbf{h}_{(C-1)}\}$, where $\mathbf{H} \in \mathbb{R}^{C \times D}$ and $\mathbf{h}_i \in \mathbb{R}^{1 \times D}$ ($i \in \{0, 1, ..., C-1\}$). We use $\mathbf{H}$ to denote graph node features of the directed graph (*i.e.*, graph topology) $\mathbf{A} \in \mathbb{R}^{C \times C}$.

### 3.2.2 GCN Refinement Block

The GCN refinement block has a learnable pooling-unpooling mechanism that progressively coarsens the original graph $\mathbf{A}_0$ into a series of coarsened graphs, *i.e.*, $\mathbf{A}_1, \mathbf{A}_2 ... \mathbf{A}_n$, and graph convolution is conducted on graphs at all different levels. Specifically, such a pooling operation is achieved by merging graph nodes, namely, via a learned matching matrix $\mathbf{M}$. Also, correlation matrices of different graphs, denoted as $\mathbf{A}_l$ [2], which are learned using `MLP` layers, which are also influenced by the GM responsible for generating the input image. This further emphasizes the significant impact of GM on the correlation graph generation process.

**Learnable Graph Pooling Layer** First, $\mathbf{A}_l \in \mathbb{R}^{m \times m}$ and $\mathbf{A}_{l+1} \in \mathbb{R}^{n \times n}$ denote directed graphs at $l$ th and $l+1$ th layers, with $m$ and $n$ ($m \geq n$) graph nodes, respectively. An assignment matrix $\mathbf{M}_l \in \mathbb{R}^{m \times n}$ converts $\mathbf{A}_l$ to $\mathbf{A}_{l+1}$ as:

$$\mathbf{A}_{l+1} = \mathbf{M}_l^T \mathbf{A}_l \mathbf{M}_l. \tag{2}$$

Also, we use $\mathbf{H}_l \in \mathbb{R}^{m \times D}$ and $\mathbf{H}_{l+1} \in \mathbb{R}^{n \times D}$ to denote graph node features of $\mathbf{A}_l$ and $\mathbf{A}_{l+1}$, respectively. Therefore, we can use $\mathbf{M}_l$ to perform the graph node aggregation operation via:

$$\mathbf{H}_{l+1} = \mathbf{M}_l^T \mathbf{H}_l. \tag{3}$$

For simplicity, we use $f_{GCN}$ to denote the mapping function imposed by a GCN block that has multiple GCN layers. Assuming the $q$-th and $l$-th graph layer are the first and last layer of the GCN block, then we have:

$$\mathbf{H}_l = f_{GCN}(\mathbf{H}_q). \tag{4}$$

Ideally, the learnable pooling operation should be dependent on the learned representation of the input image, and such representation is converted into $\mathbf{H}_l$. Therefore, we employ a trainable weight $\mathbf{W_m}$ to transform $\mathbf{H}_l$ into the assignment matrix $\mathbf{M}_l$:

$$\mathbf{M}_l = \frac{1}{1 + e^{-\alpha(\mathbf{W_m} \mathbf{H}_l)}}, \tag{5}$$

where $\alpha$ is set as $1e9$. Values of the resultant $\mathbf{M}_l$ are approximately equal to $0$ or $1$. It is worth noting that prior works [39, 40] also adopt techniques similar to Eq. 5 for making the thresholding operation differentiable.

---

[2] A $l$-th layer graph has nodes and connectivities (*e.g.*, correlations), and we use $\mathbf{A}_l$ to denote the $l$-th layer graph or only its correlations.

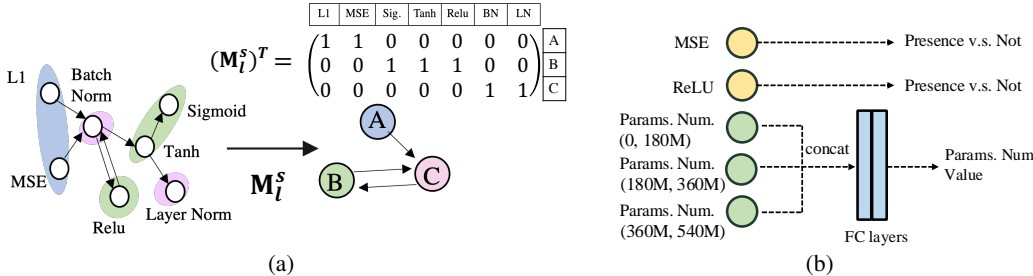

(a)                 (b)

Figure 4: (a) A toy example of the hyperparameter hierarchy assignment $\mathbf{M}_l^s$: both `L1` and `MSE` belong to the category of pixel-level loss function, so they are merged into the supernode `A`. Nonlinearity functions (*e.g.*, `ReLu` and `Tanh`) and normalization methods (*e.g.*, `Layer Norm.` and `Batch Norm.`) are merged into supernodes `B` and `C`, respectively. (b) In inference, discrete-value graph node features are used to classify if discrete hyperparameters are used in the given GM. We concatenate corresponding continuous-value node features and regress the continuous hyperparameter value.

**Learnable Unpooling Layer** We perform the graph unpooling operation that progressively restores pooled graphs to the graph at the original resolution for the graph node classification task. As shown in Fig. 2, to avoid confusion, we use $\mathbf{H}$ and $\mathbf{V}$ to represent the graph node feature on the pooling and unpooling branches, respectively. The correlation matrix on the unpooling branch is denoted by $\mathbf{A}'$,

$$\mathbf{A}'_{l-1} = \mathbf{M}_l \mathbf{A}'_l \mathbf{M}_l^T; \mathbf{V}_{l-1} = \mathbf{M}_l \mathbf{V}_l, \tag{6}$$

where $\mathbf{A}'_l$ and $\mathbf{A}'_{l-1}$ are the $l$ th and $l-1$ th layers in the unpooling branch, respectively. Finally, we use the refined feature $\mathbf{V}$ for model parsing.

**Discussion** This learnable pooling-unpooling mechanism offers three distinct advantages. First, each supernode in the coarsened graph serves as the combination of features from its children nodes, and graph convolutions on supernodes have a large receptive field for aggregating features. Secondly, the learnable correlation models hyperparameter dependencies dynamically based on generation artifacts of input image features (*e.g.*, $\mathbf{f}$). Lastly, learned correlation graphs $\mathbf{A}$ vary across different levels, helping address the over-smoothing issue commonly encountered in GCN learning [38, 45, 5].

### 3.3 Training and Inference

We jointly train our method with three objective functions. Graph node classification loss (*e.g.*, $\mathcal{L}^{graph}$) encourages each graph node feature to predict the corresponding hyperparameter label. Artifacts isolation loss (*e.g.*, $\mathcal{L}^{iso}$) helps the LGPN only parse the hyperparameters for generated images, and hyperparameter hierarchy constraints (*e.g.*, $\mathcal{L}^{hier}$) imposes hierarchical constraints among different hyperparameters while stabilizing the training.

**Training Samples** We denote a training sample as $\{\mathbf{I}, \mathbf{y}\}$, in which $\mathbf{y} = \{\mathbf{y}^d, \mathbf{y}^c, \mathbf{y}^l\} = \{y_0, y_1, ..., y_{(C-1)}\}$ is annotations of $C$ graph nodes for $G$ parsed hyperparameters as introduced in Sec. 3.1. Specifically, $y_c$ is assigned as 1 if the sample has $c$-th hyperparameter and 0 otherwise, where $c \in \{0, 1, ..., C-1\}$. Details are in Supplementary Sec. B.

**Graph Node Classification Loss** Given the image $\mathbf{I}$, we convert the refined feature $\mathbf{V}$ into the predicted score vector, denoted as $\mathbf{s} = \{s_0, s_1, ..., s_{(C-1)}\}$. We employ the sigmoid activation to retrieve the probability vector $\mathbf{p} = \{p_0, p_1, ..., p_{(C-1)}\}$, namely, $p_c = \texttt{SIGMOID}(s_c)$. Then, we have:

$$\mathcal{L}^{graph} = \sum_{c=0}^{C-1} (y_c \log p_c + (1 - y_c) \log(1 - p_c)). \tag{7}$$

**Hyperparameter Hierarchy Prediction** Fig. 4 shows that different hyperparameters can be grouped, so we define the hyperparameter hierarchy assignment $\mathbf{M}^s$ to reflect this inherent nature. More details of such the assignment are in Supplementary Sec. A. Suppose, at the $l$ th layer, we minimize the $L_2$ norm of the difference between the predicted matching matrix $\mathbf{M}_l$ and $\mathbf{M}_l^s$,

$$\mathcal{L}^{hier} = \|\mathbf{M}_l^s - \mathbf{M}_l\|_2 = \sqrt{\sum_{i,j=0} (m_{ij}^s - m_{ij})^2}. \tag{8}$$

|  |  | Method |  | Loss Function | | Dis. Archi. Para. | | Con. Archi. Para. |
|---|---|---|---|---|---|---|---|---|
|  | ID | Backbone | MP Head | F1 ↑ | Acc. ↑ | F1 ↑ | Acc. ↑ | L1 error ↓ |
| 1 | Baseline1 | ResNet-50 | MLP | 79.0 | 77.7 | 72.1 | 69.0 | 0.163 |
| 2 | Baseline2 | HR-Net | MLP | 80.7 | 81.9 | 72.2 | 70.3 | 0.149 |
| 3 | Baseline3 | ViT-B | MLP | 76.3 | 75.9 | 68.3 | 66.4 | 0.177 |
| 4 | Baseline4 | GTC | MLP | 82.5 | 80.9 | 75.7 | 70.9 | 0.135 |
| 5 | FEN-PN [2] | FEN. | Parsing Net. | 80.5 | 78.9 | 73.0 | 70.8 | 0.139 |
| 6 | LGPN | GTC | GCN refinement | **84.6** | **83.3** | **79.5** | **77.5** | **0.120** |

Table 1: We report model parsing performance on RED140, where each method has an individual ID that represents different backbones and model parsing (MP) heads. The comparison among different backbones (⬜) shows the effectiveness of the proposed GTC. *Loss Function* reports the averaged prediction performance on 10 loss functions. The averaged performance on 18 discrete architecture hyperparameters and 9 continuous hyperparameters are reported in *Dis. Archi. Para.* and *Con. Archi. Para.*, respectively. [**Bold**: best result].

**Artifacts Isolation Loss** We denote the image-level binary label as $y^{img}$ and use $p^{img}$ as the probability that $\mathbf{I}$ is a generated image. Then we have:

$$\mathcal{L}^{iso} = \sum_{i=0}^{M-1} (y^{img} \log p^{img} + (1 - y^{img}) \log(1 - p^{img})). \tag{9}$$

In summary, our joint training loss function can be written as $\mathcal{L}^{all} = \lambda_1 \mathcal{L}^{graph} + \lambda_2 \mathcal{L}^{hier} + \lambda_3 \mathcal{L}^{iso}$, where $\lambda_1$ and $\lambda_2$ equal 0 when $\mathbf{I}$ is real.

**Inference** As Fig. 4b, we use the discrete-value graph node feature to perform the binary classification to decide the presence of given hyperparameters. For the continuous architecture hyperparameter, we first concatenate $n$ corresponding node feature and train a linear layer to regress it to the estimated value. Empirically, we set $n$ as 3 and show this concatenated feature improves the robustness in predicting the continuous value (see Supplementary Tab. 7).

## 4 Experiment

### 4.1 Model Parsing

**Setup** Our experiment utilizes RED140 dataset. In RED140, each GM contains $1,000$ images, resulting in a total of $140,000$ generated images that encompass a wide range of semantics, including objects, handwritten digits, and human faces. Also, RED140 has real images on which these GMs are trained, such as CelebA [42], MNIST [14], CIFAR10 [36], ImageNet [13], facades [77], edges2shoes [77] and, apple2oranges [77]. We follow the protocol of [2]: we divide samples into 4 disjoint sets, each of which comprises different GM categories such as GAN, VAE, DM, *etc*. Next, we do leave-one-out testing, *i.e.*, train on 125 GMs from three sets, and test on GMs of the remaining set. The performance is averaged across four test sets, measured by F1 score and accuracy for discrete hyperparameters (loss function and discrete architecture hyperparameters) and L1 error for continuous architecture hyperparameters. Implementation details are in Supplementary Sec. B.

**Main Performance** We report model parsing performance in Tab. 1, where our proposed LGPN (line #6) largely outperforms previous model parsing algorithms. We first employ commonly used backbones to set up competitive model parsing baselines for a more comprehensive comparison. More formally, four baselines in lines #1—4 that use ResNet-50, ViT-B, HR-Net, and GTC as backbones with 2 layers MLP as the model parsing head, respectively. Baseline4 (line #4) achieves the best performance, which indicates that GTC is the most suitable backbone for model parsing. Specifically, Baseline4 has $1.8\%$ and $3.5\%$ higher F1 score over Baseline2 that uses HR-Net on predicting hyperparameters of loss functions and discrete architecture hyperparameters. After that, we report FEN-PN's performance, which already proves the effectiveness on the model parsing task since it has specific model parsing architectures containing a fingerprint estimate network (FEN) and a parsing network that predicts hyperparameters. Surprisingly, although FEN-PN has achieved competitive results on RED116, it only has comparable performance to Baseline2. This indicates that FEN-PN reduces its effectiveness in predicting hyperparameters of diffusion models, as RED140

| | Method | | Loss Function | | Dis. Archi. Para. | |
|---|---|---|---|---|---|---|
| | Backbone | MP Head | F1 ↑ | Acc. ↑ | F1 ↑ | Acc. ↑ |
| 1 | | MLP | 82.5 | 80.9 | 75.7 | 70.9 |
| 2 | | Stacked GCN | 83.2 | 81.3 | 76.9 | 73.8 |
| 3 | GTC | Att-GCN [61] | 83.4 | 82.7 | 78.0 | 74.5 |
| 4 | | Graph U-Net [19] | 82.2 | 82.0 | 74.8 | 70.2 |
| 5 | | GCN refinement | **84.6** | **83.3** | **79.5** | **77.5** |

(a)                   (b)

Table 2: (a) Model parsing performance comparison with different GCN variants [**Bold**: best result]. (b) The GCN refinement block improves prediction performance on continuous hyperparameters.

| Training Objectives | | | F1 score ↑ | |
|---|---|---|---|---|
| $\mathcal{L}^{graph}$ | $\mathcal{L}^{iso}$ | $\mathcal{L}^{hier}$ | Loss Fun. | Dis. Archi. Para. |
| ✔ | | | 83.7 | 77.0 |
| ✔ | ✔ | | 84.0 | 78.1 |
| ✔ | | ✔ | 83.9 | 79.0 |
| ✔ | ✔ | ✔ | **84.6** | **79.5** |

| Method | Loss Function | Dis. Archi. Para. | Con. Archi. Para. |
|---|---|---|---|
| | F1 score ↑ | | L1 error ↓ |
| FEN-PN [2] | 81.3 | 71.8 | 0.149 |
| GTC w MLP | 77.8 | 68.9 | 0.169 |
| GTC w S-GCN | 79.0 | 69.8 | 0.145 |
| LGPN | **84.1** | **74.3** | **0.130** |

(a)       (b)       (c)

Figure 5: a) Cosine similarity between generated correlation graphs (*i.e.*, $\mathbf{A}'_0$) for *unseen* GMs in one of four test sets. Each element of this matrix is the average cosine similarities of $2,000$ pairs of generated correlation graphs $\mathbf{A}'_0$ from corresponding GMs. b) The ablation on three objective functions, defined in Sec. 3.3. c) The model parsing performance on RED116 dataset. [Key: **Best**; S-GCN: stacked GCN]

contains more images from diffusion models than RED116. The complete performance comparison on RED116 is reported in Tab. 5c. Lastly, we replace MLP with the GCN refinement block, which is the full model of LGPN (line #6) and performs better on all metrics, demonstrating that the GCN refinement block indeed refines graph node features and makes a more effective model parsing head than MLP layers.

**Hyperparameter Dependency Capturing** Tab. 2a reports model parsing performance using different GCN variants as the model parsing head. Overall, the proposed GCN refinement block, which refines graph node features to better capture hyperparameter dependencies, helps achieve the best model parsing performance. By comparing lines #1 and #2, we conclude that replacing MLP layers with the GCN refinement benefits the model parsing task. After all, GCN leverages structural information of the pre-defined graph, improving the learning of correlations among different hyperparameters. Next, we use *Graph attention networks* [61] (Att-GCN) at line #3, which employs the attention mechanism to update the graph node feature. As a result, Att-GCN achieves $1.1\%$ higher F1 than stacked GCN (*e.g.*, line #2) on predicting discrete architecture parameters. Lastly, line #4 uses *Graph U-Net* [19], which has a similar pooling-unpooling mechanism to our GCN refinement block, but its pooling operation discards graph nodes in the previous layer for forming a smaller graph. Using the Graph U-Net as the model parsing head produces the worse performance on discrete architecture hyperparameters — $4.3\%$ lower than Att-GCN (#3). We believe this is because dropping graph nodes is not optimal for model parsing, in which all graph nodes represent corresponding hyperparameters and are important for the final performance. In contrast, LGPN merges children nodes into the supernode, so all node information in the previous layer remains in the smaller pooled graph, helping achieve the best performance on discrete hyperparameters in Tab. 2a. On the other hand, *continuous hyperparameter prediction* can also benefit from the GCN refinement block. In Tab. 2b, LGPN shows L1 errors of $0.147$ and $0.081$ on Layers Num. and Param. Num. respectively, whereas the model with MLP layers only achieves $0.149$ and $0.148$, respectively. This is because the GCN refinement learns the dependency between Param. Num. and Layer Num., which aligns with the observation that models with more layers typically have more parameters. Therefore, modeling such dependencies ultimately decreases the L1 error in Param. Num. prediction. Likewise, when predicting Conv. Layer Num. and Filters Num., the GCN refinement achieves $0.137$ and $0.149$ L1 error, whereas GTC with MLP layers have $0.151$ and $0.161$ L1 error.

**GM-dependent Graph** Fig. 5a shows that learned correlation graphs ($\mathbf{A}'_0$) from image pairs exhibit significant similarity when both images belong to the same *unseen* GM. This result demonstrates that our correlation graph largely depends on the GM instead of image contents, given that we have

| Method | Acc. | AUC | Pd@5% |
|---|---|---|---|
| HiFi-Net [24] | 72.3 | 75.4 | 30.4 |
| FEN-PN [2] | 83.0 | 92.4 | 61.2 |
| GTC *w* MLP | 83.9 | 92.2 | 62.5 |
| GTC *w* S-GCN | 84.3 | 94.2 | 68.6 |
| LGPN | 85.9 | 95.7 | 77.2 |

(a)

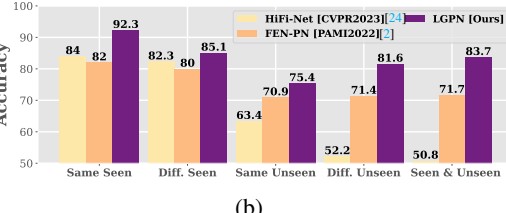

(b)

Table 3: (a) The average coordinate attack detection performance on 4-fold cross validation. (b) Test image pairs in coordinate attack detection are from one of 5 cases, bar charts from left to right: same and different GMs in seen set; same or different GMs in unseen set; one GM is from seen set and the other from unseen set. [Key: S-GCN: stacked GCN]

different contents (*e.g.*, human face and objects) in unseen GMs from each test. In addition, we empirically observe our method remains robust when using different thresholds to construct the graph, which is detailed in the Supplementary Tab. 9. However, the performance declines more when the threshold increases to 0.65, which causes the correlation graph to have very sparse connectivities, hindering the learning of hyperparameter dependencies.

**Objective Functions Analysis**  Fig. 5b shows the ablation of different training objective functions introduced in Sec. 3.3. Line #1 only optimizes the LGPN with $\mathcal{L}^{grpah}$, producing results comparable to simply stacking GCN with the attention mechanism (*e.g.*, line #3 in Tab. 2a). Lines #2 and #3 show contributions from $\mathcal{L}^{iso}$ and $\mathcal{L}^{hier}$, which improve the performance by $1.1\%$ and $2.0\%$ than only using $\mathcal{L}^{grpah}$, on predicting discrete architecture hyperparameters, respectively. This is because $\mathcal{L}^{iso}$ and $\mathcal{L}^{hier}$ make the LGPN concentrate on learning generation traces from generated images and impose the hierarchical constraints, respectively.

**RED116 Results**  Fig. 5c reports that LGPN achieves the best performance on all metrics in RED116 dataset. Interestingly, LGPN obtains $79.5\%$ F1 score on predicting discrete architecture hyperparameters in RED140 (Tab. 1), whereas only $74.3\%$ in RED116, which does not contain diffusion model generated images. We believe this is because all diffusion models share similar architectures, and such similarities make the prediction of their architecture hyperparameters easier.

### 4.2 Coordinate Attack Detection

We evaluate the proposed LGPN on coordinated attacks detection [2], which aims to classify whether two fake images are generated from the same GM or not. This is achieved by computing the cosine similarity between predicted hyperparameters from given images. Specifically, we evaluate coordinated attack detection on RED140 in a 4-fold cross-validation. In each fold, the train set has 125 GMs, and the test set has 30 GMs where 15 GMs are exclusive (unseen) from the train set and 15 GMs are seen in the train set. We generate $89,000$ training image pairs from train-set GMs for training $1,000$ image pairs for validation. We generate $25,000$ test image pairs from test-set GMs, and the average of the 4 folds is used as the final result. For the measurement, we use accuracy, AUC, and detection probability at a fixed false alarm rate (Pd@FAR) *e.g.*, Pd@5% as metrics. Specifically, aside from FEN-PN [2], we also compare with the recent work HiFi-Net [24] that show SoTA results in attributing different forgery methods. Specifically, we train HiFi-Net to classify 125 GMs and take learned features from the last fully-connected layer for coordinated attack detection. The performance is reported in Tab. 3a, which demonstrates that our proposed method surpasses both prior works. We observe the HiFi-Net performs much worse on AUC than the model parsing baseline (*e.g.*, GTC *w* S-GCN) and FEN-PN. Furthermore, Tab. 3b shows the FEN-PN and LGPN perform comparably as HiFi-Net on seen GMs (first two bar charts), yet much better than HiFi-Net when images are generated by unseen GMs (last three bar charts). Lastly, LGPN has a better Pd@5% performance than other methods.

### 4.3 Capturing Generation Traces

To study GTC's ability to identify generation traces, we adopt it for CNN-generated image detection and image attribution. For a fair comparison, *no* model parsing dataset is used for the pre-training.

**CNN-generated Image Detection**  We append fully-connected layers at the end of GTC to obtain a binary detector that distinguishes CNN-generated images from real ones (detailed in supplementary

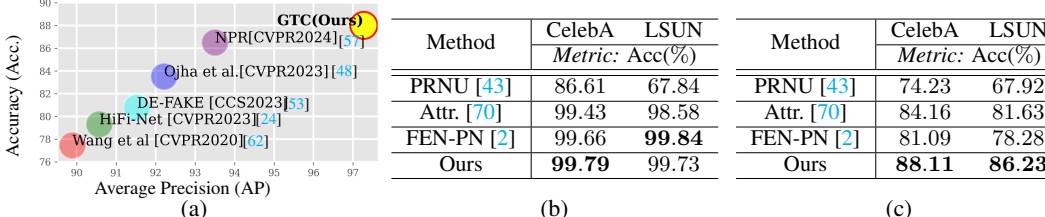

Figure 6: (a) CNN-generated image detection performance. (b) and (c) report image attribution performance in two different protocols.

Fig. 8). We follow the experiment setup from prior works [62, 48, 57], which trains the model on images generated by ProGAN [32], and test it on images generated by 11 unseen forgery methods, using average precision (AP) and accuracy for the measurement. Fig. 6a reports that our method achieves premium detection performance compared to prior methods. The second-best method, NPR [57], focuses on learning local up-sampling artifacts from pixels, helping detect images from unseen GMs. Instead, GTC's high-resolution representation more effectively exploits both local and global traces left from generation processes, obtaining a better performance.

**Image Attribution** Tab. 6b and Tab. 6c report the image attribution performance in two different protocols. Specifically, we define the protocol 1 based on the previous work [2], which trains methods on $100,000$ real and $100,000$ images generated from four different GMs (*e.g.*, SNGAN, MMDGAN, CRAMERGAN, and ProGAN), conducting a five-way classification (*i.e.*, 4 GMs and real). In protocol 2, we add two more generative methods (*e.g.*, styleGANv2, styleGANv3), resulting in a more challenging task: a 7-way classification task, classifying whether the image is real samples or generated by which one of 6 GMs. As depicted in Supplementary Fig. 8, we apply fully-connected layers on the top of GTC, which leverages the final representation of the generation trace for the multi-category classification task, *e.g.*, image attribution. Our proposed method achieves the best image attribution performance on CelebA and competitive results on LSUN, indicating that GTC has a promising ability to capture the generation trace.

## 5 Conclusion

In this study, our focus is *model parsing*, which predicts pre-defined hyperparameters of a GM given an input image. We propose a novel method that incorporates a learnable pooling-unpooling mechanism devised for the model parsing task. This mechanism serves multiple purposes: modeling GM-dependent hyperparameter dependencies, expanding the receptive field of graph convolution, and mitigating the over-smoothing issue in GCN learning. In addition, we provide the Generation Trace Capturing network to capture generation artifacts, which proves effective in two different image forensic applications: CNN-generated image detection and coordinated attacks detection.

**Limitation** We empirically observe two limitations in our proposed method, both of which can be interesting directions for future research. First, while our model parsing approach delivers excellent performance on the RED140 dataset, it is worth exploring its effectiveness on specific GMs that fall outside our dataset scope. This investigation would provide valuable insights into the generalizability of our method to a broader range of GMs. Secondly, we formulate the model parsing task as a closed-set classification problem, which limits its ability to predict undefined hyperparameters, *e.g.*, LeakyReLU. One interesting solution could be adding a few new graph nodes for undefined hyperparameters while keeping original graph nodes — the learned dependency between graph nodes representing ReLU and LeakyReLU should be high.

**Broader Impact** We strongly advocate for the machine learning and computer vision community to actively work towards mitigating potential negative societal implications of research. It is possible that generated face images used in training could leak the identity information of subjects who have not provided consent forms. We shall strive to work on real face imagery whose collection is reviewed by an Institutional Review Board (IRB).

**Acknowledge** This work was supported by the Defense Advanced Research Projects Agency (DARPA) under Agreement No. HR00112090131 to Xiaoming Liu at Michigan State University.

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

In this supplementary, we provide:

◇ Predictable hyperparameters introduction.

◇ Training and implementation details.

◇ Additional results of model parsing and CNN-generated image detection.

◇ The construction of RED140 dataset.

◇ Hyperparameter ground truth and the model parsing performance for each GM

## A  Predictable Hyperparameters Introduction

We investigate 37 hyperparameters that exhibit the predictability according to Asnani *et al.* [2]. These hyperparameters are categorized into three groups: (1) Loss Function (Tab. 4), (2) Discrete Architecture Hyperparameters (Tab. 5), (3) Continuous Architecture Hyperparameters (Tab. 6). We report our proposed method performance of parsing hyperparameters in these three groups via Fig. 7a, Fig. 7b, and Fig. 7c, respectively. Moreover, in the main paper's Eq. 8 and Fig. 4a, we employ the assignment hierarchy $\mathbf{M}^s$ to group different hyperparameters together, which supervises the learning of the matching matrix $\mathbf{M}$. The construction of such the $\mathbf{M}^s$ is also based on Tab. 4, 5, and 6, which not only define three coarse-level categories, but also fine-grained categories such as pixel-level objective (loss) function (*e.g.*, L1 and MSE) in Tab. 4, and normalization methods (*e.g.*, ReLu and Tanh) as well as nonlinearity functions (*e.g.*, Layer Norm. and Batch Norm.) in Tab. 5.

## B  Training and Implementation Details

**Training Details**  Given the directed graph $\mathbf{A} \in \mathbb{R}^{C \times C}$, which contains $C$ graph nodes. We empirically set $C$ as 55, as mentioned in the main paper Sec. 3.3. In the training, LGPN takes the given image $\mathbf{I}$ and output the refined feature $\mathbf{V} \in \mathbb{R}^{55 \times D}$, which contains learned features for each graph node, namely, $\mathbf{V} = \{\mathbf{v}_0, \mathbf{v}_1, ..., \mathbf{v}_{54}\}$. As a matter of fact, we can view $\mathbf{V}$ as three separate sections: $\mathbf{V}^l = \{\mathbf{v}_0, \mathbf{v}_1, ..., \mathbf{v}_9\}$, $\mathbf{V}^d = \{\mathbf{v}_{10}, \mathbf{v}_{11}, ..., \mathbf{v}_{27}\}$, and $\mathbf{V}^c = \{\mathbf{v}_{28}, \mathbf{v}_{29}, ..., \mathbf{v}_{54}\}$, which denote learned features for graph nodes of 10 loss functions (*e.g.*, L1 and MSE), 18 discrete architecture hyperparameter (*e.g.*, Batch Norm. and ReLU), and 9 continuous architecture hyperparameter (*i.e.*, Parameter Num.), respectively. Note $\mathbf{V}^c$ represents learned features of 9 continuous architecture hyperparameters because each continuous hyperparameter is represented by 3 graph nodes, as illustrated in Fig. 1c of the main paper. Furthermore, via Eq. 7 in the main paper, we use $\mathbf{V}$ to obtain the corresponding probability score $\mathbf{p} = \{p_0, p_1, ..., p_{54}\}$ for each graph node. Similar to $\mathbf{V}$, this $\mathbf{p}$ can be viewed as three sections: $\mathbf{p}^l \in \mathbb{R}^{10}$, $\mathbf{p}^d \in \mathbb{R}^{18}$ and $\mathbf{p}^c \in \mathbb{R}^{27}$ for loss functions, discrete architecture hyperparameters, continuous architecture hyperparameters, respectively. In the end, we use $\mathbf{p}$ to help optimize LGPN via the graph node classification loss (Eq. 7). After the training converges, we further apply individual fully connected layers on the top of frozen learned features of continuous architecture hyperparameters (*e.g.*, $\mathbf{V}^c$). via minimizing the $\mathcal{L}_1$ distance between predicted and ground truth value.

In the inference (the main paper Fig. 4b), for loss function and discrete architecture hyperparameters, we use output probabilities (*e.g.*, $\mathbf{p}^l$ and $\mathbf{p}^d$) of discrete value graph nodes, for the binary "used v.s. not" classification. For the continuous architecture hyperparameters, we first concatenate learned features of corresponding graph nodes. We utilize such a concatenated feature with pre-trained, fully connected layers to estimate the continuous hyperparameter value.

**Model Parsing Implementation Details**  Denote the output feature from the Generation Trace Capturing Network as $\mathbf{f} \in \mathbb{R}^{2048}$. To transform $\mathbf{f}$ into a set of features $\mathbf{H} = \{\mathbf{h}_0, \mathbf{h}_1, ..., \mathbf{h}_{54}\}$ for the 55 graph nodes, 55 independent linear layers ($\Theta$) are employed. Each feature $\mathbf{h}_i$ is of dimension $\mathbb{R}^{512}$. The $\mathbf{H}$ is fed to the GCN refinement block, which contains 5 GCN blocks, each of which has 2 stacked GCN layers. In other words, the GCN refinement block has 10 layers in total. We use the correlation graph $\mathbf{A} \in \mathbb{R}^{55 \times 55}$ (Fig. 10) to capture the hyperparameter dependency and during the training the LGPN pools $\mathbf{A}$ into $\mathbf{A}_1 \in \mathbb{R}^{18 \times 18}$ and $\mathbf{A}_2 \in \mathbb{R}^{6 \times 6}$ as the Fig. 2 of the main paper. The LGPN is implemented using the PyTorch framework. During training, a learning rate of 3e-2 is used. The training is performed with a batch size of 400, where 200 images are generated by various GMs and 200 images are real.

Table 4: Loss Function types used by all GMs. We group the 10 loss functions into three categories. We use the binary representation to indicate the presence of each loss type in training the respective GM.

| Category | Loss Function |
|---|---|
| Pixel-level | $L_1$ |
| | $L_2$ |
| | Mean squared error (MSE) |
| | Maximum mean discrepancy (MMD) |
| | Least squares (LS) |
| Discriminator | Wasserstein loss for GAN (WGAN) |
| | Kullback–Leibler (KL) divergence |
| | Adversarial |
| | Hinge |
| Classification | Cross-entropy (CE) |

Table 5: Discrete Architecture Hyperparameters used by all GMs. We group the 18 discrete architecture hyperparameters into 6 categories. We use the binary representation to indicate the presence of each hyperparameter type in training the respective GM.

| Category | Discrete Architecture Hyperparameters |
|---|---|
| Normalization | Batch Normalization |
| | Instance Normalization |
| | Adaptive Instance Normalization |
| | Group Normalization |
| Nonlinearity in the Last Layer | ReLU |
| | Tanh |
| | Leaky_ReLU |
| | Sigmoid |
| | SiLU |
| Nonlinearity in the Last Block | ELU |
| | ReLU |
| | Leaky_ReLU |
| | Sigmoid |
| | SiLU |
| Up-sampling | Nearest Neighbour Up-sampling |
| | Deconvolution |
| Skip Connection | Feature used |
| Down-sampling | Feature used |

Table 6: Continuous Architecture Hyperparameters used by all GMs, where "[" denotes inclusive and "(" denotes exclusive intervals. We report the range for 9 continuous hyperparameters.

| Category | Range | Discrete Architecture Hyperparameters |
|---|---|---|
| Layer Number | $(0\text{—}717]$ | Layers Number |
| | $[0\text{—}289]$ | Convolution Layer Number |
| | $[0\text{—}185]$ | Fully-connected Layer Number |
| | $[0\text{—}46]$ | Pooling Layer Number |
| | $[0\text{—}235]$ | Normalization Layer Number |
| | $(0\text{—}20]$ | Layer Number per Block |
| Unit Number | $(0\text{—}8,365]$ | Filter Number |
| | $(0\text{—}155]$ | Block Number |
| | $(0\text{—}56,008,488]$ | Parameter Number |

**Implementation Details for Detection and Attributions** We validate GTC's effectiveness in capturing the generation trace in Fig. 6. Specifically, Fig. 8 shows the detailed implementation. We employ FC layers to convert output feature $\mathbf{f} \in \mathbb{R}^{2048}$ to $\mathbf{f}_{det.} \in \mathbb{R}^2$ and $\mathbf{f}_{att.} \in \mathbb{R}^5$ for CNN-generated image detection and image attribution respectively.

## C  Additional Results

We report detailed performance on RED116 via Tab. 8, demonstrating that our proposed LGPN achieves the best performance on both datasets. Also, in Fig. 5a of the main paper, we visualize

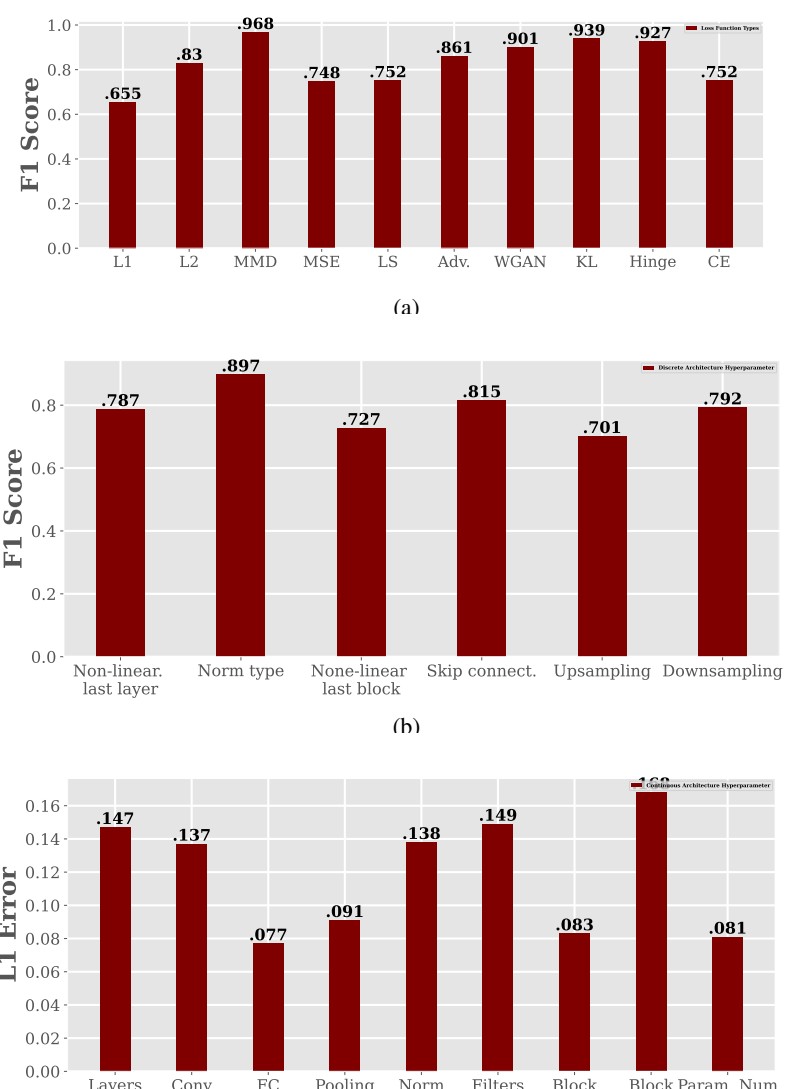

Figure 7: (a) The F1 score on the loss function reveals that `MMD` and `KL` are two easiest loss functions to predict. (b) The F1 score on the discrete architecture hyperparameters demonstrates that predicting these hyperparameters is more challenging than predicting the loss function. This finding aligns with the empirical results reported in the previous work [2]. (c) The L1 error on the continuous architecture hyperparameters indicates that it is challenging to predict `Block Num.` and `Filter Num.`.

| $n$ Value | 2 | 3 | 4 | 5 | 6 |
|---|---|---|---|---|---|
| L1 Error | 0.123 | 0.120 | 0.124 | 0.132 | 0.130 |

Table 7: Using different $n$ graph nodes for the continuous hyperparameter regression.

the correlation graph similarities among different GMs in the first test set. In this section, we offer a similar visualization (*e.g.,* Fig. 9 of the supplementary) for other test sets. In the main paper's Sec 3.3, we use $n$ graph nodes for each continuous hyperparameter and $n$ is set as 3. Tab. 7 shows the advantage of choice, which shows the lowest $\mathcal{L}_1$ regression error is achieved when $n$ is 3.

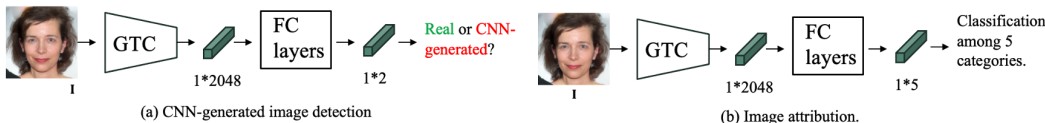

(a) CNN-generated image detection

(b) Image attribution.

Figure 8: We construct simple classifiers based on GTC. Then, we train these two classifiers for CNN-generated image detection and image attribution, respectively. Please note that GTC only leverages ImageNet pre-trained weights as the initialization, same as the previous method [62]. For a fair comparison, no model parsing datasets such as RED116 and RED140 are used for pre-training.

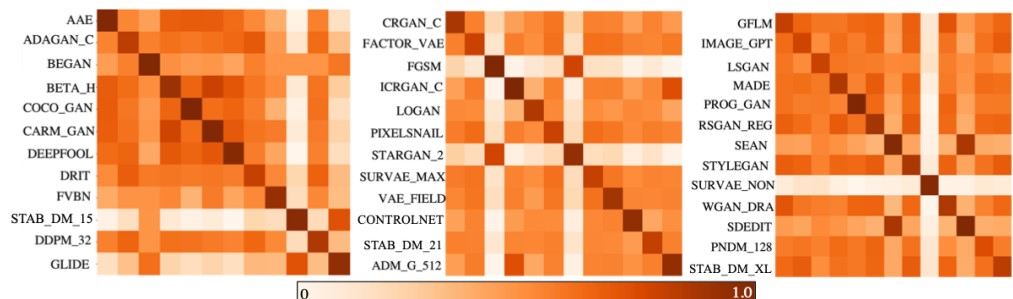

Figure 9: Each element of these two matrices is the average cosine similarities of $2,000$ pairs of generated correlation graphs $\mathbf{A}'_0$ from corresponding GMs in the second, third and forth test sets.

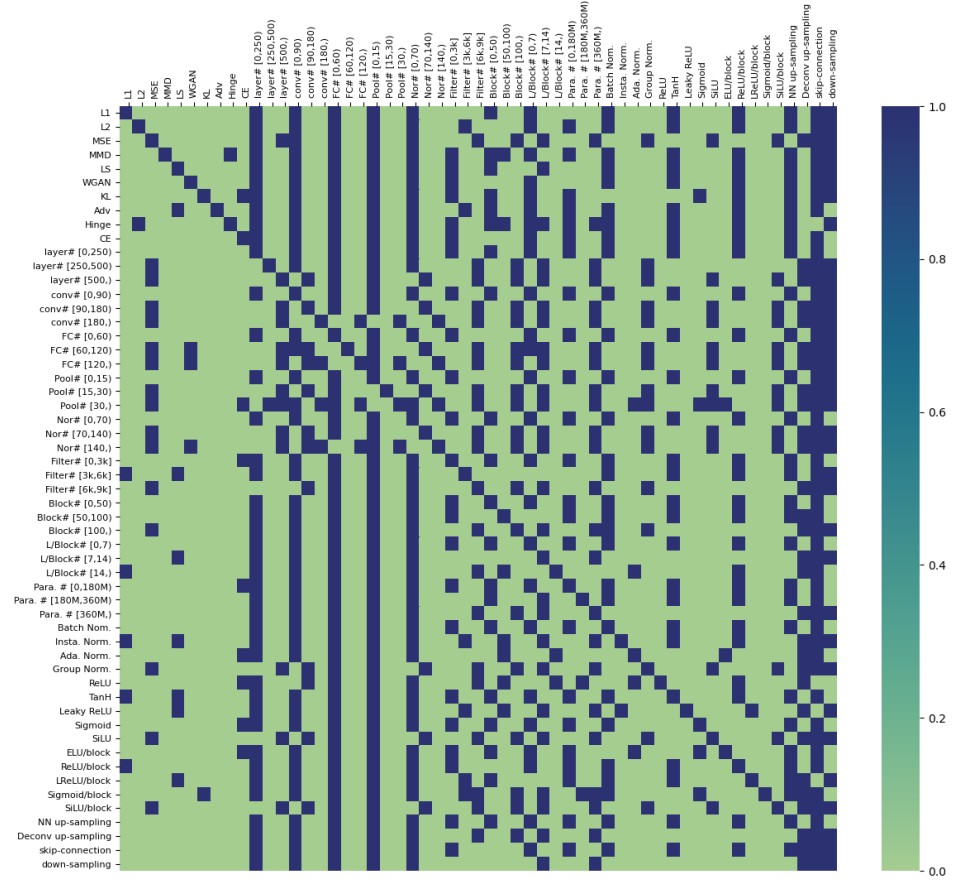

Figure 10: The initial correlation graph $\mathbf{A}$ that we construct based on the probability table in Sec. 3.1 of the main paper. The optimum threshold we use is $0.45$.

| Method | Loss Function | | Dis. Archi. Para. | | Con. Archi. Para. |
|---|---|---|---|---|---|
| | F1 ↑ | Acc. ↑ | F1 ↑ | Acc. ↑ | L1 error ↓ |
| Random GT [2] | 0.636 | 0.716 | 0.529 | 0.575 | 0.184 |
| FEN-PN [2] | 0.813 | 0.792 | 0.718 | 0.706 | 0.149 |
| FEN-PN* [2] | 0.801 | 0.811 | 0.701 | 0.708 | 0.146 |
| GTC *w* MLP | 0.778 | 0.801 | 0.689 | 0.701 | 0.169 |
| GTC *w* Stacked -GCN | 0.790 | 0.831 | 0.698 | 0.720 | 0.145 |
| LGPN | **0.841** | **0.833** | **0.727** | **0.755** | **0.130** |

Table 8: The model parsing performance on RED116. In the last row, our proposed LGPN that contains GTC and GCN Refinement block, which achieves the best model parsing performance in all metrics. [**Key**: GCN refine.: GCN refinement block; **Bold**: best.].

| Threshold | Loss Function | Dis. Archi. Para. |
|---|---|---|
| | F1/Accuracy ↑ | |
| 0.35 | 84.0/83.0 | 79.2/77.0 |
| 0.45 | **84.6/83.3** | **79.5/77.5** |
| 0.55 | 84.5/82.8 | 78.9/77.0 |
| 0.65 | 82.7/82.5 | 77.0/74.5 |

Table 9: More parsing performance with different thresholds constructing the correlation graph **A**.

| Method | Test GM | Train GM | Con. Archi. Para. $L_1$ error ↓ | Dis. Archi. Para. F1 ↑ | Loss function F1 ↑ |
|---|---|---|---|---|---|
| FEN-PN | Face | Face | $0.139 \pm 0.042$ | $0.729 \pm 0.106$ | $0.788 \pm 0.146$ |
| Ours | Face | Face | $0.112 \pm 0.028$ | $0.786 \pm 0.116$ | $0.801 \pm 0.134$ |
| FEN-PN | Face | Non-Face | $0.213 \pm 0.066$ | $0.688 \pm 0.125$ | $0.759 \pm 0.1$ |
| Ours | Face | Non-Face | $0.139 \pm 0.063$ | $0.694 \pm 0.117$ | $0.771 \pm 0.2$ |
| FEN-PN | Face | Full | $0.118 \pm 0.046$ | $0.712 \pm 0.129$ | $0.833 \pm 0.136$ |
| Ours | Face | Full | $0.099 \pm 0.044$ | $0.745 \pm 0.099$ | $0.840 \pm 0.123$ |
| FEN-PN | Non-Face | Non-Face | $0.118 \pm 0.021$ | $0.794 \pm 0.11$ | $0.864 \pm 0.094$ |
| Ours | Non-Face | Face | $0.116 \pm 0.016$ | $0.810 \pm 0.102$ | $0.870 \pm 0.092$ |
| FEN-PN | Non-Face | Face | $0.125 \pm 0.031$ | $0.667 \pm 0.099$ | $0.858 \pm 0.115$ |
| Ours | Non-Face | Non-Face | $0.100 \pm 0.027$ | $0.692 \pm 0.101$ | $0.882 \pm 0.112$ |
| FEN-PN | Non-Face | Full | $0.082 \pm 0.045$ | $0.832 \pm 0.046$ | $0.886 \pm 0.061$ |
| Ours | Non-Face | Full | $0.080 \pm 0.042$ | $0.844 \pm 0.032$ | $0.901 \pm 0.021$ |

Table 10: Performance comparison across different face and non-face GMs.

| Method | Continuous type | | | | Discrete type | |
|---|---|---|---|---|---|---|
| | $L_1$ error ↓ | P-value ↓ | Corr. coef. ↑ | Coef. of det. ↑ | F1 score ↑ | Accuracy ↑ |
| Random ground-truth | $0.184 \pm 0.019$ | $0.006 \pm 0.001$ | $0.261 \pm 0.181$ | $0.315 \pm 0.095$ | $0.529 \pm 0.078$ | $0.575 \pm 0.097$ |
| Mean/mode | $0.164 \pm 0.011$ | $0.035 \pm 0.005$ | $0.326 \pm 0.112$ | $0.467 \pm 0.115$ | $0.612 \pm 0.048$ | $0.604 \pm 0.046$ |
| No fingerprint | $0.170 \pm 0.035$ | $0.017 \pm 0.004$ | $0.738 \pm 0.014$ | $0.605 \pm 0.152$ | $0.700 \pm 0.032$ | $0.663 \pm 0.104$ |
| Using one parser | $0.161 \pm 0.028$ | $0.032 \pm 0.002$ | $0.226 \pm 0.030$ | $0.512 \pm 0.116$ | $0.607 \pm 0.034$ | $0.593 \pm 0.104$ |
| FEN-PN | $0.149 \pm 0.019$ | $0.022 \pm 0.007$ | $0.744 \pm 0.098$ | $0.612 \pm 0.161$ | $0.718 \pm 0.036$ | $0.706 \pm 0.040$ |
| Ours | $0.130 \pm 0.011$ | N/A | $0.833 \pm 0.098$ | $0.732 \pm 0.177$ | $0.743 \pm 0.033$ | $0.755 \pm 0.030$ |

Table 11: Performance of architecture hyperparameters prediction. We use $L_1$ error, p-value, correlation coefficient, and coefficient of determination for continuous type parameters. For discrete architecture hyperparameters, we use the F1 score and classification accuracy. The first value is the standard deviation across sets, while the second one is across samples. The p-value is estimated for every ours-baseline pair. [KEYS: corr.: correlation, coef.: coefficient, det.: determination]

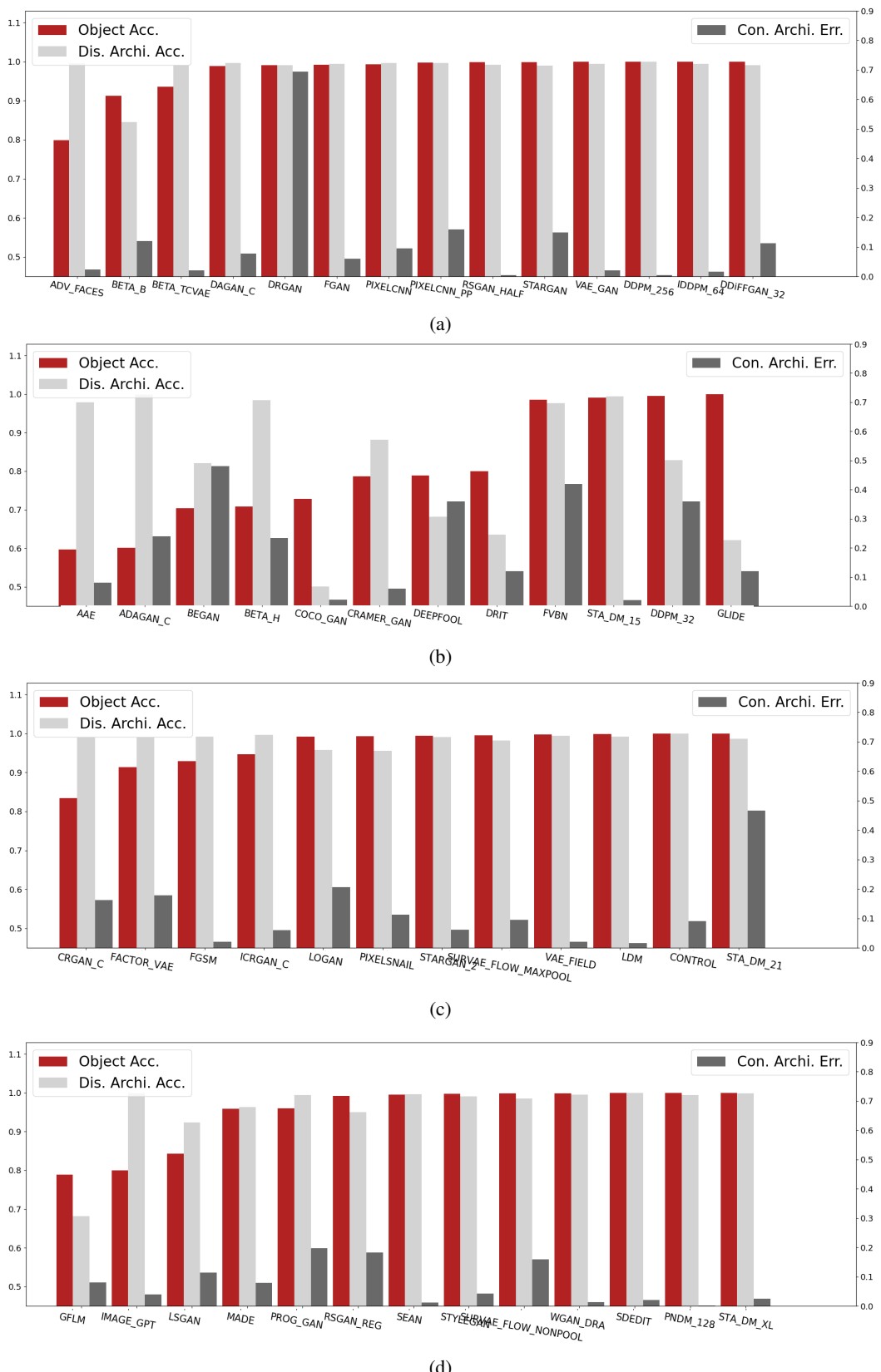

Figure 11: We report detailed model parsing results on different GMs in each test set. These results include loss function and discrete architecture hyperparameter prediction accuracy, as well as the L1 error on the continuous architecture hyperparameter prediction. Specifically, (a), (b), (c), and (d) are the performance for GMs in the 1st, 2nd, 3rd, and 4th test sets, respectively.

Table 12: Test sets used for evaluation. Each set contains generative models from GAN, DM, VAE, AR (Auto-Regressive), AA (Adversarial Attack), and NF (Normalizing Flow). All test sets contain face and non-face in the image content. [Keys: R means GM is used in the test set of RED116 but is not used in RED140.]

| Set 1 | Set 2 | Set 3 | Set 4 |
|---|---|---|---|
| ADV_FACES | AAE | BICYCLE_GAN (R) | GFLM |
| BETA_B | ADAGAN_C | BIGGAN_512 (R) | IMAGE_GPT |
| BETA_TCVAE | BEGAN | CRGAN_C | LSGAN |
| BIGGAN_128 (R) | BETA_H | FACTOR_VAE | MADE |
| DAGAN_C | BIGGAN_256 (R) | FGSM | PIX2PIX (R) |
| DRGAN | COCOGAN | ICRGAN_C | PROG_GAN |
| FGAN | CRAMERGAN | LOGAN | RSGAN_REG |
| PIXEL_CNN | DEEPFOOL | MUNIT (R) | SEAN |
| PIXEL_CNN++ | DRIT | PIXEL_SNAIL | STYLE_GAN |
| RSGAN_HALF | FAST_PIXEL(R) | STARGAN_2 | SURVAE_FLOW_NONPOOL |
| STARGAN | FVBN | SURVAE_FLOW_MAXPOOL | WGAN_DRA |
| VAEGAN | SRFLOW (R) | VAE_FIELD | YLG (R) |
| DDPM_256 | ADM_G_64 | LDM | ADM_G_128 |
| IDDPM_64 | DDPM_32 | CONTROLNET | STABLE_DM_XL |
| Denoise_GAN_32 | GLIDE | STABLE_DM_15 | SEDdit |

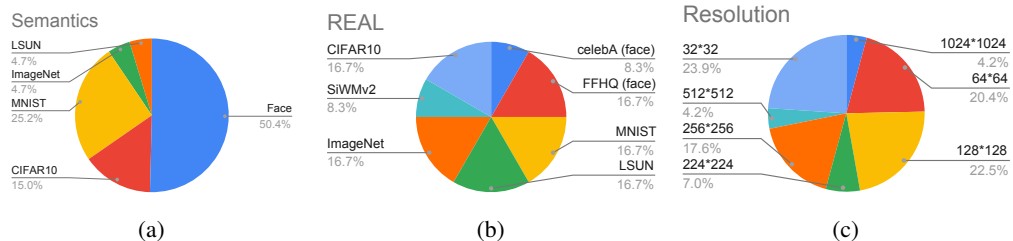

(a)  (b)  (c)

Figure 12: RED140 statistics. (a) The dataset is trained on various image contents or semantics. (b) The real-image category contains many real-image datasets that GMs are trained on [33, 32, 13, 69, 25, 36, 37]. (c) The GM has various image resolutions.

## D   RED140 Dataset

In this section, we provide an overview of the RED140 dataset, which is used for both model parsing and coordinated attack detection. Note that, for the experiment reported in Tab. **??** of the supplementary, we follow the test sets defined in RED116 [2]. When we construct RED140, we use images from ImageNet, FFHQ, CelebHQ, CIFAR10, and LSUN as the real-images category of RED140. We exclude GM that is not trained in the real-image category of RED140. In addition, both RED116 and RED140 contain various image content and resolution, and the details about RED140 are uncovered in Fig. 12 of the supplementary. For test sets (Tab. 12 of the supplementary), we follow the dataset partition of RED116, excluding the GMs that are trained on real images, which RED140 does have. For example, JFT-300M is used to train BigGAN, so we remove `BIGGAN_128`, `BIGGAN_256` and `BIGGAN_512` in the first, second, and third test sets.

## E   GM Hyperparameter Ground Truth

In this section, we report the ground truth vector of different hyperparameters of each GM contained in the RED140. Specifically, Tab. 13 and Tab. 14 report the loss function ground truth vector for each GM. Tab. 15 and Tab. 16 report the discrete architecture hyperparameter ground truth for each GM. Tab. 17 and Tab. 18 report the continuous architecture hyperparameter ground truth for each GM. The detailed model parsing performance on each GM is reported in the supplementary's Fig. 11.

Table 13: Ground truth Loss Function feature vector used for prediction of loss type for all GMs. The loss function ground truth is in (Tab. 4).

| GM | $L_1$ | $L_2$ | MSE | MMD | LS | WGAN | KL | Adversarial | Hinge | CE |
|---|---|---|---|---|---|---|---|---|---|---|
| AAE | 1 | 0 | 0 | 0 | 0 | 0 | 0 | 0 | 0 | 1 |
| ACGAN | 1 | 0 | 0 | 0 | 0 | 0 | 0 | 0 | 0 | 1 |
| ADAGAN_C | 0 | 0 | 0 | 0 | 1 | 0 | 0 | 0 | 0 | 1 |
| ADAGAN_P | 0 | 0 | 0 | 0 | 1 | 0 | 0 | 0 | 0 | 0 |
| ADM_G_128 | 0 | 0 | 1 | 0 | 0 | 0 | 0 | 0 | 0 | 0 |
| ADM_G_256 | 0 | 0 | 1 | 0 | 0 | 0 | 0 | 0 | 0 | 0 |
| ADV_FACES | 1 | 0 | 1 | 0 | 1 | 0 | 0 | 0 | 0 | 0 |
| ALAE | 0 | 0 | 1 | 0 | 1 | 0 | 0 | 0 | 0 | 0 |
| BEGAN | 1 | 0 | 0 | 0 | 0 | 0 | 0 | 0 | 0 | 0 |
| BETA_B | 0 | 0 | 0 | 0 | 0 | 0 | 1 | 0 | 0 | 1 |
| BETA_H | 0 | 0 | 0 | 0 | 0 | 0 | 1 | 0 | 0 | 1 |
| BETA_TCVAE | 1 | 0 | 0 | 0 | 0 | 0 | 1 | 0 | 0 | 1 |
| BGAN | 0 | 0 | 0 | 0 | 1 | 0 | 0 | 0 | 0 | 1 |
| BICYCLE_GAN | 1 | 0 | 1 | 0 | 0 | 0 | 1 | 0 | 0 | 0 |
| BIGGAN_128 | 1 | 0 | 0 | 0 | 0 | 0 | 0 | 0 | 0 | 0 |
| BIGGAN_256 | 1 | 0 | 0 | 0 | 0 | 0 | 0 | 0 | 0 | 0 |
| BIGGAN_512 | 1 | 0 | 0 | 0 | 0 | 0 | 0 | 0 | 0 | 0 |
| Blended_DM | 0 | 0 | 1 | 0 | 0 | 0 | 0 | 0 | 0 | 0 |
| CADGAN | 0 | 0 | 0 | 1 | 0 | 0 | 0 | 0 | 0 | 0 |
| CCGAN | 0 | 0 | 0 | 0 | 1 | 0 | 0 | 1 | 0 | 0 |
| CGAN | 0 | 0 | 1 | 0 | 1 | 0 | 0 | 0 | 0 | 0 |
| CLIPDM | 0 | 0 | 1 | 0 | 0 | 0 | 0 | 0 | 0 | 0 |
| COCO_GAN | 1 | 1 | 0 | 0 | 0 | 1 | 0 | 0 | 1 | 0 |
| COGAN | 0 | 0 | 0 | 0 | 1 | 0 | 0 | 0 | 0 | 0 |
| COLOUR_GAN | 1 | 0 | 0 | 0 | 1 | 0 | 0 | 0 | 0 | 0 |
| CONT_ENC | 0 | 1 | 0 | 0 | 1 | 0 | 0 | 0 | 0 | 0 |
| CONTRAGAN | 1 | 0 | 0 | 0 | 0 | 0 | 0 | 1 | 0 | 1 |
| CONTROLNET | 0 | 0 | 1 | 0 | 0 | 0 | 0 | 0 | 0 | 0 |
| COUNCIL_GAN | 1 | 0 | 1 | 0 | 1 | 0 | 0 | 0 | 0 | 0 |
| CRAMER_GAN | 0 | 0 | 0 | 0 | 0 | 1 | 0 | 0 | 0 | 0 |
| CRGAN_C | 1 | 1 | 0 | 0 | 0 | 0 | 0 | 0 | 0 | 1 |
| CRGAN_P | 1 | 1 | 0 | 0 | 0 | 0 | 0 | 0 | 0 | 0 |
| CYCLEGAN | 1 | 0 | 0 | 0 | 1 | 0 | 0 | 0 | 0 | 0 |
| DAGAN_C | 1 | 0 | 0 | 0 | 0 | 0 | 0 | 0 | 0 | 1 |
| DAGAN_P | 1 | 0 | 0 | 0 | 0 | 0 | 0 | 0 | 0 | 0 |
| DCGAN | 0 | 0 | 0 | 0 | 0 | 0 | 0 | 0 | 0 | 1 |
| DDPM_32 | 0 | 0 | 1 | 0 | 0 | 0 | 0 | 0 | 0 | 0 |
| DDPM_256 | 0 | 0 | 1 | 0 | 0 | 0 | 0 | 0 | 0 | 0 |
| DDiFFGAN_32 | 0 | 0 | 1 | 0 | 0 | 1 | 0 | 0 | 0 | 0 |
| DEEPFOOL | 1 | 1 | 0 | 0 | 0 | 0 | 0 | 0 | 0 | 0 |
| DFCVAE | 0 | 1 | 0 | 0 | 0 | 0 | 1 | 0 | 0 | 1 |
| DIFFAE_256 | 0 | 0 | 1 | 0 | 0 | 0 | 0 | 0 | 0 | 0 |
| DIFFAE_LATENT | 0 | 0 | 1 | 0 | 0 | 0 | 0 | 0 | 0 | 0 |
| DIFF-ProGAN | 0 | 0 | 0 | 0 | 0 | 0 | 1 | 0 | 0 | 0 |
| DIFF-StyleGAN | 0 | 0 | 0 | 0 | 0 | 0 | 1 | 0 | 0 | 0 |
| DIFF-ISGEN | 0 | 0 | 0 | 0 | 0 | 0 | 1 | 0 | 0 | 0 |
| DISCOGAN | 1 | 0 | 0 | 0 | 1 | 0 | 0 | 0 | 0 | 0 |
| DRGAN | 0 | 0 | 0 | 0 | 1 | 0 | 0 | 0 | 0 | 1 |
| DRIT | 1 | 0 | 0 | 0 | 1 | 0 | 0 | 0 | 0 | 1 |
| DUALGAN | 1 | 0 | 0 | 0 | 0 | 1 | 0 | 0 | 0 | 0 |
| EBGAN | 0 | 1 | 0 | 0 | 1 | 0 | 0 | 1 | 1 | 0 |
| ESRGAN | 1 | 0 | 0 | 0 | 1 | 0 | 0 | 0 | 0 | 0 |
| FACTOR_VAE | 1 | 0 | 0 | 0 | 0 | 0 | 1 | 0 | 0 | 1 |
| Fast pixel | 0 | 0 | 0 | 0 | 0 | 0 | 0 | 0 | 0 | 1 |
| FFGAN | 1 | 1 | 0 | 0 | 1 | 0 | 0 | 0 | 0 | 1 |
| FGAN | 0 | 0 | 0 | 0 | 1 | 0 | 0 | 1 | 0 | 0 |
| FGAN_KL | 1 | 0 | 0 | 0 | 0 | 0 | 0 | 0 | 0 | 0 |
| FGAN_NEYMAN | 0 | 1 | 0 | 0 | 0 | 0 | 0 | 0 | 0 | 0 |
| FGAN_PEARSON | 0 | 0 | 1 | 0 | 0 | 0 | 0 | 0 | 1 | 0 |
| FGSM | 0 | 0 | 0 | 0 | 1 | 0 | 0 | 0 | 0 | 0 |
| FPGAN | 1 | 1 | 0 | 0 | 1 | 0 | 0 | 0 | 0 | 1 |
| FSGAN | 1 | 0 | 0 | 0 | 1 | 0 | 0 | 0 | 0 | 1 |
| FVBN | 0 | 0 | 0 | 0 | 0 | 0 | 0 | 0 | 0 | 1 |
| GAN_ANIME | 1 | 1 | 0 | 0 | 0 | 1 | 0 | 0 | 1 | 0 |
| Gated_pixel_cnn | 0 | 0 | 0 | 0 | 0 | 0 | 0 | 0 | 0 | 1 |
| GDWCT | 1 | 0 | 1 | 0 | 0 | 0 | 0 | 0 | 1 | 0 |
| GFLM | 0 | 0 | 1 | 0 | 0 | 0 | 0 | 0 | 0 | 1 |
| GGAN | 1 | 0 | 0 | 0 | 0 | 0 | 0 | 0 | 0 | 0 |
| GLIDE | 0 | 0 | 1 | 0 | 0 | 0 | 0 | 0 | 0 | 0 |
| ICRGAN_C | 1 | 1 | 0 | 0 | 0 | 0 | 0 | 0 | 0 | 1 |
| ICRGAN_P | 1 | 1 | 0 | 0 | 0 | 0 | 0 | 0 | 0 | 0 |

Table 14: Ground truth Loss Function feature vector used for prediction of loss type for all GMs. The loss function ground truth is in (Tab. 4).

| GM | $L_1$ | $L_2$ | MSE | MMD | LS | WGAN | KL | Adversarial | Hinge | CE |
|---|---|---|---|---|---|---|---|---|---|---|
| IDDPM_32 | 0 | 0 | 1 | 0 | 0 | 0 | 0 | 0 | 0 | 0 |
| IDDPM_64 | 0 | 0 | 1 | 0 | 0 | 0 | 0 | 0 | 0 | 0 |
| IDDPM_256 | 0 | 0 | 1 | 0 | 0 | 0 | 0 | 0 | 0 | 0 |
| ILVER_256 | 0 | 0 | 1 | 0 | 0 | 0 | 0 | 0 | 0 | 0 |
| Image_GPT | 0 | 0 | 0 | 0 | 0 | 0 | 0 | 0 | 0 | 1 |
| INFOGAN | 0 | 0 | 1 | 0 | 1 | 0 | 0 | 0 | 0 | 1 |
| LAPGAN | 0 | 0 | 0 | 0 | 1 | 0 | 0 | 0 | 0 | 0 |
| LDM | 0 | 0 | 1 | 0 | 0 | 0 | 0 | 0 | 0 | 0 |
| LDM_CON | 0 | 0 | 1 | 0 | 0 | 1 | 0 | 0 | 0 | 0 |
| Lmconv | 0 | 0 | 0 | 0 | 0 | 0 | 0 | 0 | 0 | 1 |
| LOGAN | 1 | 1 | 0 | 0 | 0 | 0 | 0 | 1 | 0 | 0 |
| LSGAN | 0 | 0 | 1 | 0 | 0 | 0 | 0 | 0 | 1 | 0 |
| MADE | 0 | 0 | 0 | 0 | 0 | 0 | 0 | 0 | 0 | 1 |
| MAGAN | 0 | 0 | 1 | 0 | 0 | 0 | 0 | 0 | 0 | 0 |
| MEMGAN | 0 | 0 | 0 | 0 | 1 | 0 | 0 | 0 | 0 | 0 |
| MMD_GAN | 1 | 0 | 0 | 1 | 0 | 0 | 0 | 0 | 0 | 0 |
| MRGAN | 0 | 0 | 1 | 0 | 1 | 0 | 0 | 0 | 0 | 0 |
| MSG_STYLE_GAN | 0 | 0 | 0 | 0 | 1 | 0 | 0 | 0 | 0 | 0 |
| MUNIT | 1 | 0 | 0 | 0 | 1 | 0 | 0 | 0 | 0 | 0 |
| NADE | 0 | 0 | 0 | 0 | 0 | 0 | 0 | 0 | 0 | 1 |
| OCFGAN | 0 | 0 | 0 | 1 | 0 | 0 | 0 | 0 | 1 | 0 |
| PGD | 1 | 1 | 0 | 0 | 0 | 0 | 0 | 0 | 0 | 0 |
| PIX2PIX | 1 | 0 | 0 | 0 | 1 | 0 | 0 | 0 | 0 | 0 |
| PixelCNN | 0 | 0 | 0 | 0 | 0 | 0 | 0 | 0 | 0 | 1 |
| PixelCNN++ | 0 | 0 | 0 | 0 | 0 | 0 | 0 | 0 | 0 | 1 |
| PIXELDA | 0 | 0 | 0 | 0 | 1 | 0 | 0 | 0 | 1 | 1 |
| PixelSnail | 0 | 0 | 0 | 0 | 0 | 0 | 0 | 0 | 0 | 1 |
| PNDM_32 | 0 | 0 | 1 | 0 | 0 | 0 | 0 | 0 | 0 | 0 |
| PNDM_256 | 0 | 0 | 1 | 0 | 0 | 0 | 0 | 0 | 0 | 0 |
| PROG_GAN | 0 | 0 | 0 | 0 | 0 | 1 | 0 | 0 | 1 | 0 |
| RGAN | 0 | 0 | 0 | 0 | 0 | 1 | 0 | 0 | 0 | 0 |
| RSGAN_HALF | 0 | 0 | 0 | 0 | 0 | 0 | 0 | 0 | 0 | 1 |
| RSGAN_QUAR | 0 | 0 | 0 | 0 | 0 | 0 | 0 | 0 | 0 | 1 |
| RSGAN_REG | 0 | 0 | 0 | 0 | 0 | 0 | 0 | 0 | 0 | 1 |
| RSGAN_RES_BOT | 0 | 0 | 0 | 0 | 0 | 0 | 0 | 0 | 0 | 1 |
| RSGAN_RES_HALF | 0 | 0 | 0 | 0 | 0 | 0 | 0 | 0 | 0 | 1 |
| RSGAN_RES_QUAR | 0 | 0 | 0 | 0 | 0 | 0 | 0 | 0 | 0 | 1 |
| RSGAN_RES_REG | 0 | 0 | 0 | 0 | 0 | 0 | 0 | 0 | 0 | 1 |
| SAGAN | 0 | 0 | 0 | 0 | 1 | 0 | 0 | 0 | 0 | 0 |
| SCOREDIFF_256 | 0 | 0 | 1 | 0 | 0 | 0 | 0 | 0 | 0 | 0 |
| SDEdit_256 | 0 | 0 | 1 | 0 | 0 | 0 | 0 | 0 | 0 | 0 |
| SEAN | 1 | 0 | 0 | 0 | 1 | 0 | 0 | 0 | 0 | 0 |
| SEMANTIC | 0 | 1 | 0 | 0 | 1 | 0 | 0 | 0 | 0 | 0 |
| SGAN | 0 | 0 | 0 | 0 | 1 | 0 | 0 | 0 | 0 | 1 |
| SNGAN | 0 | 0 | 0 | 0 | 1 | 0 | 0 | 1 | 0 | 0 |
| SOFT_GAN | 0 | 0 | 0 | 0 | 1 | 0 | 0 | 0 | 0 | 0 |
| SRFLOW | 1 | 0 | 0 | 0 | 0 | 0 | 0 | 0 | 0 | 1 |
| SRRNET | 0 | 1 | 1 | 0 | 1 | 0 | 0 | 0 | 0 | 1 |
| STANDARD_VAE | 0 | 0 | 0 | 0 | 0 | 0 | 1 | 0 | 0 | 1 |
| STARGAN | 1 | 0 | 0 | 0 | 1 | 0 | 0 | 0 | 0 | 1 |
| STARGAN_2 | 1 | 0 | 0 | 0 | 1 | 0 | 0 | 0 | 0 | 0 |
| STA_DM_15 | 0 | 0 | 1 | 0 | 0 | 0 | 0 | 0 | 0 | 0 |
| STA_DM_21 | 0 | 0 | 1 | 0 | 0 | 0 | 0 | 0 | 0 | 0 |
| STA_DM_XL | 0 | 0 | 1 | 0 | 0 | 0 | 0 | 0 | 0 | 0 |
| STGAN | 1 | 0 | 0 | 0 | 1 | 1 | 0 | 0 | 0 | 0 |
| STYLEGAN | 0 | 1 | 0 | 0 | 0 | 1 | 0 | 0 | 0 | 0 |
| STYLEGAN_2 | 0 | 1 | 0 | 0 | 1 | 0 | 0 | 0 | 1 | 0 |
| STYLEGAN2_ADA | 0 | 1 | 0 | 1 | 1 | 0 | 0 | 0 | 1 | 0 |
| SURVAE_FLOW_MAXPOOL | 0 | 0 | 0 | 0 | 0 | 0 | 1 | 0 | 0 | 1 |
| SURVAE_FLOW_NONPOOL | 0 | 0 | 0 | 0 | 0 | 0 | 1 | 0 | 0 | 1 |
| TPGAN | 1 | 0 | 0 | 0 | 0 | 1 | 0 | 0 | 0 | 0 |
| UGAN | 0 | 0 | 0 | 0 | 1 | 0 | 0 | 0 | 0 | 0 |
| UNIT | 0 | 0 | 0 | 0 | 1 | 0 | 1 | 0 | 0 | 0 |
| VAE_field | 0 | 0 | 0 | 0 | 0 | 0 | 1 | 0 | 0 | 1 |
| VAE_flow | 0 | 0 | 0 | 0 | 0 | 0 | 1 | 0 | 0 | 1 |
| VAEGAN | 1 | 0 | 0 | 0 | 1 | 0 | 1 | 0 | 0 | 0 |
| VDVAE | 0 | 0 | 0 | 0 | 0 | 0 | 1 | 0 | 0 | 1 |
| WGAN | 0 | 0 | 0 | 0 | 0 | 1 | 0 | 0 | 0 | 0 |
| WGAN_DRA | 0 | 0 | 1 | 0 | 0 | 1 | 0 | 0 | 0 | 0 |
| WGAN_WC | 0 | 0 | 0 | 0 | 0 | 1 | 0 | 0 | 0 | 0 |
| WGANGP | 0 | 1 | 0 | 0 | 0 | 1 | 0 | 0 | 0 | 0 |
| YLG | 0 | 0 | 0 | 0 | 0 | 1 | 0 | 0 | 0 | 0 |

Table 15: Ground truth feature vector used for prediction of Discrete Architecture Hyperparameters for all GMs. The discrete architecture hyperparameter ground truth is defined in (Tab. 5). **A — D** are Batch Norm., Instance Norm., Adaptive Instance Norm., and Group Norm., respectively. **E — I** are non-linearity in the last layer, and they are ReLU, Tanh, Leaky_ReLu, Sigmoid, and SiLU. **J — N** are non-linearity in the last block, and they are ELU, ReLU, Leaky_ReLu, Sigmoid, and SiLU. **O** and **P** are Nearest Neighbour and Deconvolution Upsampling. **Q** and **L** are Skip Connection and Downsampling.

| GM | A | B | C | D | E | F | G | H | I | J | K | L | M | N | O | P | Q | L |
|---|---|---|---|---|---|---|---|---|---|---|---|---|---|---|---|---|---|---|
| AAE | 1 | 0 | 0 | 0 | 0 | 1 | 0 | 0 | 0 | 1 | 0 | 0 | 0 | 0 | 1 | 0 | 1 | 0 |
| ACGAN | 1 | 0 | 0 | 0 | 0 | 1 | 0 | 0 | 0 | 0 | 1 | 0 | 0 | 0 | 1 | 0 | 1 | 0 |
| ADAGAN_C | 1 | 0 | 0 | 0 | 0 | 1 | 0 | 0 | 0 | 0 | 1 | 0 | 0 | 0 | 1 | 0 | 1 | 0 |
| ADAGAN_P | 1 | 0 | 0 | 0 | 1 | 0 | 0 | 0 | 0 | 0 | 1 | 0 | 0 | 0 | 1 | 0 | 1 | 0 |
| ADM_G_128 | 0 | 0 | 1 | 0 | 0 | 0 | 0 | 1 | 0 | 0 | 0 | 0 | 0 | 1 | 1 | 0 | 1 | 1 |
| ADM_G_256 | 0 | 0 | 1 | 0 | 0 | 0 | 0 | 1 | 0 | 0 | 0 | 0 | 0 | 1 | 1 | 0 | 1 | 1 |
| ADV_FACES | 0 | 1 | 0 | 0 | 0 | 1 | 0 | 0 | 0 | 0 | 1 | 0 | 0 | 0 | 1 | 0 | 1 | 0 |
| ALAE | 0 | 1 | 0 | 0 | 0 | 0 | 1 | 0 | 0 | 0 | 0 | 1 | 0 | 0 | 0 | 1 | 0 | 1 |
| BEGAN | 1 | 0 | 0 | 0 | 0 | 1 | 0 | 0 | 0 | 1 | 0 | 0 | 0 | 0 | 1 | 0 | 0 | 0 |
| BETA_B | 0 | 0 | 0 | 0 | 0 | 0 | 0 | 1 | 0 | 0 | 1 | 0 | 0 | 0 | 1 | 0 | 1 | 1 |
| BETA_H | 0 | 0 | 0 | 0 | 0 | 0 | 0 | 1 | 0 | 0 | 1 | 0 | 0 | 0 | 1 | 0 | 1 | 1 |
| BETA_TCVAE | 0 | 0 | 0 | 0 | 0 | 0 | 0 | 1 | 0 | 0 | 1 | 0 | 0 | 0 | 1 | 0 | 1 | 1 |
| BGAN | 1 | 0 | 0 | 0 | 0 | 1 | 0 | 0 | 0 | 0 | 0 | 1 | 0 | 0 | 1 | 0 | 0 | 0 |
| BICYCLE_GAN | 1 | 0 | 0 | 0 | 0 | 1 | 0 | 0 | 0 | 0 | 1 | 0 | 0 | 0 | 1 | 0 | 0 | 0 |
| BIGGAN_128 | 1 | 0 | 0 | 0 | 0 | 1 | 0 | 0 | 0 | 0 | 1 | 0 | 0 | 0 | 0 | 1 | 1 | 1 |
| BIGGAN_256 | 1 | 0 | 0 | 0 | 0 | 1 | 0 | 0 | 0 | 0 | 1 | 0 | 0 | 0 | 0 | 1 | 1 | 1 |
| BIGGAN_512 | 1 | 0 | 0 | 0 | 0 | 1 | 0 | 0 | 0 | 0 | 1 | 0 | 0 | 0 | 0 | 1 | 1 | 1 |
| Blended_DM | 0 | 0 | 0 | 1 | 0 | 0 | 0 | 0 | 1 | 0 | 0 | 0 | 0 | 1 | 0 | 1 | 1 | 1 |
| CADGAN | 1 | 0 | 0 | 0 | 0 | 1 | 0 | 0 | 0 | 0 | 1 | 0 | 0 | 0 | 1 | 0 | 1 | 1 |
| CCGAN | 1 | 0 | 0 | 0 | 0 | 1 | 0 | 0 | 0 | 0 | 1 | 0 | 0 | 0 | 0 | 1 | 1 | 1 |
| CGAN | 1 | 0 | 0 | 0 | 0 | 1 | 0 | 0 | 0 | 0 | 0 | 1 | 0 | 0 | 1 | 0 | 0 | 0 |
| CLIPDM | 0 | 0 | 0 | 1 | 0 | 0 | 0 | 0 | 0 | 0 | 0 | 0 | 0 | 0 | 0 | 1 | 1 | 1 |
| COCO_GAN | 1 | 0 | 0 | 0 | 0 | 1 | 0 | 0 | 0 | 0 | 1 | 0 | 0 | 0 | 1 | 0 | 0 | 0 |
| COGAN | 1 | 0 | 0 | 0 | 0 | 1 | 0 | 0 | 0 | 0 | 0 | 1 | 0 | 0 | 1 | 0 | 1 | 1 |
| COLOUR_GAN | 1 | 0 | 0 | 0 | 0 | 1 | 0 | 0 | 0 | 0 | 1 | 0 | 0 | 0 | 1 | 0 | 1 | 1 |
| CONT_ENC | 1 | 0 | 0 | 0 | 0 | 1 | 0 | 0 | 0 | 0 | 0 | 1 | 0 | 0 | 1 | 0 | 1 | 1 |
| CONTRAGAN | 1 | 0 | 0 | 0 | 0 | 1 | 0 | 0 | 0 | 0 | 1 | 0 | 0 | 0 | 1 | 0 | 1 | 0 |
| CONTROLNET | 0 | 0 | 0 | 1 | 0 | 0 | 0 | 1 | 0 | 0 | 1 | 0 | 0 | 0 | 1 | 0 | 1 | 1 |
| COUNCIL_GAN | 0 | 1 | 0 | 0 | 0 | 1 | 0 | 0 | 0 | 0 | 1 | 0 | 0 | 0 | 1 | 0 | 1 | 0 |
| CRAMER_GAN | 1 | 0 | 0 | 0 | 0 | 1 | 0 | 0 | 0 | 0 | 1 | 0 | 0 | 0 | 1 | 0 | 1 | 0 |
| CRGAN_C | 1 | 0 | 0 | 0 | 0 | 1 | 0 | 0 | 0 | 0 | 1 | 0 | 0 | 0 | 1 | 0 | 1 | 0 |
| CRGAN_P | 1 | 0 | 0 | 0 | 0 | 1 | 0 | 0 | 0 | 0 | 1 | 0 | 0 | 0 | 1 | 0 | 1 | 0 |
| CYCLEGAN | 0 | 1 | 0 | 0 | 0 | 1 | 0 | 0 | 0 | 0 | 1 | 0 | 0 | 0 | 0 | 1 | 1 | 1 |
| DAGAN_C | 1 | 0 | 0 | 0 | 0 | 1 | 0 | 0 | 0 | 0 | 1 | 0 | 0 | 0 | 1 | 0 | 1 | 0 |
| DAGAN_P | 1 | 0 | 0 | 0 | 0 | 1 | 0 | 0 | 0 | 0 | 1 | 0 | 0 | 0 | 1 | 0 | 1 | 0 |
| DCGAN | 1 | 0 | 0 | 0 | 0 | 1 | 0 | 0 | 0 | 0 | 1 | 0 | 0 | 0 | 1 | 0 | 1 | 0 |
| DDiFFGAN_32 | 1 | 0 | 0 | 0 | 0 | 0 | 0 | 1 | 0 | 0 | 0 | 0 | 0 | 1 | 1 | 0 | 1 | 1 |
| DDPM_32 | 1 | 0 | 0 | 0 | 0 | 0 | 0 | 1 | 0 | 0 | 1 | 0 | 0 | 0 | 1 | 0 | 1 | 1 |
| DDPM_256 | 1 | 0 | 0 | 0 | 0 | 0 | 0 | 1 | 0 | 0 | 1 | 0 | 0 | 0 | 1 | 0 | 1 | 1 |
| DEEPFOOL | 0 | 0 | 1 | 0 | 1 | 0 | 0 | 0 | 0 | 0 | 1 | 0 | 0 | 0 | 0 | 1 | 0 | 0 |
| DFCVAE | 1 | 0 | 0 | 0 | 0 | 0 | 0 | 1 | 0 | 0 | 0 | 1 | 0 | 0 | 1 | 0 | 0 | 1 |
| DIFF_ISGEN | 0 | 0 | 0 | 0 | 0 | 0 | 1 | 0 | 0 | 0 | 0 | 0 | 1 | 0 | 1 | 0 | 0 | 0 |
| DIFF_PGAN | 0 | 0 | 0 | 0 | 0 | 0 | 1 | 0 | 0 | 0 | 0 | 1 | 0 | 1 | 0 | 0 | 0 | 0 |
| DIFF_SGAN | 0 | 0 | 0 | 0 | 0 | 0 | 1 | 0 | 0 | 0 | 0 | 1 | 0 | 1 | 0 | 0 | 0 | 0 |
| DIFFAE | 1 | 0 | 0 | 0 | 0 | 0 | 0 | 1 | 0 | 0 | 0 | 0 | 0 | 1 | 1 | 0 | 1 | 1 |
| DIFFAE_LATENT | 1 | 0 | 0 | 0 | 0 | 0 | 0 | 1 | 0 | 0 | 0 | 0 | 0 | 1 | 1 | 0 | 1 | 1 |
| DISCOGAN | 0 | 1 | 0 | 0 | 0 | 1 | 0 | 0 | 0 | 0 | 0 | 1 | 0 | 0 | 0 | 1 | 1 | 1 |
| DRGAN | 1 | 0 | 0 | 0 | 0 | 1 | 0 | 0 | 0 | 1 | 0 | 0 | 0 | 0 | 1 | 0 | 1 | 1 |
| DRIT | 0 | 1 | 0 | 0 | 0 | 1 | 0 | 0 | 0 | 0 | 1 | 0 | 0 | 0 | 0 | 1 | 1 | 1 |
| DUALGAN | 1 | 0 | 0 | 0 | 0 | 1 | 0 | 0 | 0 | 0 | 1 | 0 | 0 | 0 | 1 | 0 | 0 | 0 |
| EBGAN | 1 | 0 | 0 | 0 | 0 | 1 | 0 | 0 | 0 | 0 | 0 | 1 | 0 | 0 | 1 | 0 | 0 | 1 |
| ESRGAN | 0 | 0 | 1 | 0 | 0 | 0 | 1 | 0 | 0 | 0 | 0 | 1 | 0 | 0 | 0 | 1 | 0 | 0 |
| FACTOR_VAE | 0 | 0 | 0 | 0 | 0 | 0 | 0 | 1 | 0 | 0 | 1 | 0 | 0 | 0 | 1 | 0 | 1 | 1 |
| FASTPIXEL | 1 | 0 | 0 | 0 | 0 | 0 | 0 | 1 | 0 | 1 | 0 | 0 | 0 | 0 | 1 | 0 | 1 | 0 |
| FFGAN | 1 | 0 | 0 | 0 | 0 | 0 | 0 | 1 | 0 | 0 | 0 | 0 | 0 | 0 | 0 | 1 | 1 | 1 |
| FGAN | 1 | 0 | 0 | 0 | 0 | 0 | 0 | 1 | 0 | 0 | 1 | 0 | 0 | 0 | 1 | 0 | 1 | 0 |
| FGAN_KL | 1 | 0 | 0 | 0 | 0 | 0 | 0 | 1 | 0 | 0 | 1 | 0 | 0 | 0 | 1 | 0 | 1 | 0 |
| FGAN_NEYMAN | 1 | 0 | 0 | 0 | 0 | 0 | 0 | 1 | 0 | 0 | 1 | 0 | 0 | 0 | 1 | 0 | 1 | 0 |
| FGAN_PEARSON | 1 | 0 | 0 | 0 | 0 | 0 | 0 | 1 | 0 | 0 | 1 | 0 | 0 | 0 | 1 | 0 | 1 | 0 |
| FGSM | 0 | 0 | 1 | 0 | 1 | 0 | 0 | 0 | 0 | 0 | 1 | 0 | 0 | 0 | 0 | 1 | 0 | 0 |
| FPGAN | 0 | 1 | 0 | 0 | 0 | 1 | 0 | 0 | 0 | 0 | 1 | 0 | 0 | 0 | 1 | 0 | 0 | 1 |
| FSGAN | 1 | 0 | 0 | 0 | 1 | 0 | 0 | 0 | 0 | 0 | 1 | 0 | 0 | 0 | 0 | 1 | 1 | 1 |
| FVBN | 0 | 0 | 1 | 0 | 0 | 0 | 0 | 1 | 0 | 1 | 0 | 0 | 0 | 0 | 1 | 0 | 1 | 0 |
| GAN_ANIME | 0 | 0 | 0 | 0 | 0 | 1 | 0 | 0 | 0 | 0 | 1 | 0 | 0 | 0 | 1 | 0 | 1 | 1 |
| GATED_PIXEL_CNN | 0 | 0 | 1 | 0 | 0 | 0 | 0 | 1 | 0 | 0 | 0 | 1 | 0 | 0 | 0 | 1 | 1 | 0 |
| GDWCT | 0 | 1 | 0 | 0 | 0 | 1 | 0 | 0 | 0 | 0 | 1 | 0 | 0 | 0 | 1 | 0 | 0 | 1 |
| GFLM | 0 | 0 | 1 | 0 | 1 | 0 | 0 | 0 | 0 | 0 | 1 | 0 | 0 | 0 | 0 | 1 | 0 | 0 |
| GGAN | 1 | 0 | 0 | 0 | 0 | 1 | 0 | 0 | 0 | 0 | 1 | 0 | 0 | 0 | 1 | 0 | 1 | 1 |
| GLIDE | 1 | 0 | 0 | 0 | 0 | 0 | 0 | 1 | 0 | 0 | 0 | 0 | 0 | 1 | 1 | 0 | 1 | 1 |

Table 16: Ground truth feature vector used for prediction of Discrete Architecture Hyperparameters for all GMs. The discrete architecture hyperparameter ground truth is defined in (Tab. 5). **A — D** are Batch Norm., Instance Norm., Adaptive Instance Norm., and Group Norm., respectively. **E — I** are non-linearity in the last layer and they are ReLU, Tanh, Leaky_ReLu, Sigmoid, and SiLU. **J — N** are non-linearity in the last block and they are ELU, ReLU, Leaky_ReLu, Sigmoid, and SiLU. **O** and **P** are Nearest Neighbour and Deconvolution Upsampling. **Q** and **L** are Skip Connection and Downsampling.

| GM | A | B | C | D | E | F | G | H | I | J | K | L | M | N | O | P | Q | L |
|---|---|---|---|---|---|---|---|---|---|---|---|---|---|---|---|---|---|---|
| ICRGAN_C | 1 | 0 | 0 | 0 | 0 | 1 | 0 | 0 | 0 | 0 | 1 | 0 | 0 | 0 | 1 | 0 | 1 | 0 |
| ICRGAN_P | 1 | 0 | 0 | 0 | 0 | 1 | 0 | 0 | 0 | 0 | 1 | 0 | 0 | 0 | 1 | 0 | 1 | 0 |
| IDDPM_32 | 0 | 0 | 1 | 0 | 0 | 0 | 0 | 1 | 0 | 0 | 0 | 0 | 0 | 1 | 1 | 0 | 1 | 1 |
| IDDPM_64 | 0 | 0 | 1 | 0 | 0 | 0 | 0 | 1 | 0 | 0 | 0 | 0 | 0 | 1 | 1 | 0 | 1 | 1 |
| IDDPM_256 | 0 | 0 | 1 | 0 | 0 | 0 | 0 | 1 | 0 | 0 | 0 | 0 | 0 | 1 | 1 | 0 | 1 | 1 |
| IMAGE_GPT | 1 | 0 | 0 | 0 | 0 | 0 | 0 | 1 | 0 | 0 | 0 | 1 | 0 | 0 | 0 | 1 | 1 | 1 |
| INFOGAN | 1 | 0 | 0 | 0 | 0 | 1 | 0 | 0 | 0 | 0 | 0 | 1 | 0 | 0 | 1 | 0 | 0 | 1 |
| ILVER_256 | 0 | 0 | 0 | 1 | 0 | 0 | 0 | 0 | 0 | 0 | 0 | 0 | 0 | 0 | 0 | 1 | 1 | 1 |
| LAPGAN | 0 | 0 | 1 | 0 | 0 | 1 | 0 | 0 | 0 | 0 | 1 | 0 | 0 | 0 | 0 | 1 | 1 | 0 |
| LDM | 1 | 0 | 0 | 0 | 0 | 0 | 0 | 1 | 0 | 0 | 0 | 0 | 0 | 1 | 1 | 0 | 1 | 1 |
| LDM_CON | 1 | 0 | 0 | 0 | 0 | 0 | 0 | 1 | 0 | 0 | 0 | 0 | 0 | 1 | 1 | 0 | 1 | 1 |
| LMCONV | 0 | 0 | 1 | 0 | 0 | 0 | 0 | 1 | 0 | 1 | 0 | 0 | 0 | 0 | 1 | 1 | 1 | 1 |
| LOGAN | 1 | 0 | 0 | 0 | 0 | 1 | 0 | 0 | 0 | 0 | 1 | 0 | 0 | 0 | 1 | 0 | 1 | 0 |
| LSGAN | 1 | 0 | 0 | 0 | 0 | 1 | 0 | 0 | 0 | 0 | 1 | 0 | 0 | 0 | 1 | 0 | 0 | 0 |
| MADE | 0 | 0 | 1 | 0 | 0 | 0 | 0 | 1 | 0 | 1 | 0 | 0 | 0 | 0 | 1 | 0 | 1 | 0 |
| MAGAN | 1 | 0 | 0 | 0 | 0 | 1 | 0 | 0 | 0 | 0 | 1 | 0 | 0 | 0 | 1 | 0 | 1 | 0 |
| MEMGAN | 1 | 0 | 0 | 0 | 0 | 1 | 0 | 0 | 0 | 0 | 1 | 0 | 0 | 0 | 1 | 0 | 1 | 0 |
| MMD_GAN | 1 | 0 | 0 | 0 | 0 | 1 | 0 | 0 | 0 | 0 | 1 | 0 | 0 | 0 | 1 | 0 | 1 | 0 |
| MRGAN | 1 | 0 | 0 | 0 | 0 | 1 | 0 | 0 | 0 | 0 | 1 | 0 | 0 | 0 | 1 | 0 | 1 | 0 |
| MSG_STYLE_GAN | 0 | 1 | 0 | 0 | 0 | 0 | 1 | 0 | 0 | 0 | 0 | 1 | 0 | 0 | 0 | 1 | 0 | 1 |
| MUNIT | 0 | 1 | 0 | 0 | 1 | 0 | 0 | 0 | 0 | 0 | 0 | 1 | 0 | 0 | 0 | 1 | 1 | 1 |
| NADE | 0 | 0 | 1 | 0 | 0 | 0 | 0 | 1 | 0 | 1 | 0 | 0 | 0 | 0 | 1 | 0 | 1 | 0 |
| OCFGAN | 1 | 0 | 0 | 0 | 0 | 1 | 0 | 0 | 0 | 0 | 1 | 0 | 0 | 0 | 1 | 0 | 1 | 0 |
| PGD | 0 | 0 | 1 | 0 | 1 | 0 | 0 | 0 | 0 | 0 | 1 | 0 | 0 | 0 | 0 | 1 | 0 | 0 |
| PIX2PIX | 0 | 1 | 0 | 0 | 0 | 1 | 0 | 0 | 0 | 0 | 0 | 1 | 0 | 0 | 0 | 1 | 1 | 1 |
| PIXELCNN | 1 | 0 | 0 | 0 | 0 | 0 | 0 | 1 | 0 | 1 | 0 | 0 | 0 | 0 | 1 | 0 | 1 | 0 |
| PIXELCNN_PP | 0 | 0 | 1 | 0 | 0 | 0 | 0 | 1 | 0 | 1 | 0 | 0 | 0 | 0 | 1 | 1 | 1 | 1 |
| PIXELDA | 1 | 0 | 0 | 0 | 0 | 1 | 0 | 0 | 0 | 0 | 1 | 0 | 0 | 0 | 0 | 1 | 0 | 0 |
| PIXELSNAIL | 0 | 0 | 1 | 0 | 1 | 0 | 0 | 0 | 0 | 0 | 0 | 0 | 1 | 0 | 1 | 0 | 1 | 0 |
| PNDM_32 | 0 | 0 | 0 | 1 | 0 | 0 | 0 | 0 | 1 | 0 | 0 | 0 | 0 | 1 | 0 | 1 | 1 | 1 |
| PNDM_256 | 0 | 0 | 0 | 1 | 0 | 0 | 0 | 0 | 1 | 0 | 0 | 0 | 0 | 1 | 0 | 1 | 1 | 1 |
| PROG_GAN | 1 | 0 | 0 | 0 | 0 | 0 | 0 | 1 | 0 | 0 | 0 | 1 | 0 | 1 | 0 | 0 | 0 | 1 |
| RGAN | 1 | 0 | 0 | 0 | 0 | 1 | 0 | 0 | 0 | 0 | 1 | 0 | 0 | 1 | 0 | 0 | 0 | 1 |
| RSGAN_HALF | 1 | 0 | 0 | 0 | 0 | 1 | 0 | 0 | 0 | 0 | 1 | 0 | 0 | 0 | 1 | 0 | 1 | 0 |
| RSGAN_QUAR | 1 | 0 | 0 | 0 | 0 | 1 | 0 | 0 | 0 | 0 | 1 | 0 | 0 | 0 | 1 | 0 | 1 | 0 |
| RSGAN_REG | 1 | 0 | 0 | 0 | 0 | 1 | 0 | 0 | 0 | 0 | 1 | 0 | 0 | 0 | 1 | 0 | 1 | 0 |
| RSGAN_RES_BOT | 1 | 0 | 0 | 0 | 0 | 1 | 0 | 0 | 0 | 0 | 1 | 0 | 0 | 0 | 0 | 1 | 1 | 0 |
| RSGAN_RES_HALF | 1 | 0 | 0 | 0 | 0 | 1 | 0 | 0 | 0 | 0 | 1 | 0 | 0 | 0 | 0 | 1 | 1 | 0 |
| RSGAN_RES_QUAR | 1 | 0 | 0 | 0 | 0 | 1 | 0 | 0 | 0 | 0 | 1 | 0 | 0 | 0 | 0 | 1 | 1 | 0 |
| RSGAN_RES_REG | 1 | 0 | 0 | 0 | 0 | 1 | 0 | 0 | 0 | 0 | 1 | 0 | 0 | 0 | 0 | 1 | 1 | 0 |
| SAGAN | 1 | 0 | 0 | 0 | 0 | 1 | 0 | 0 | 0 | 0 | 0 | 1 | 0 | 0 | 1 | 0 | 0 | 0 |
| SCOREDIFF_256 | 0 | 0 | 0 | 1 | 0 | 0 | 0 | 0 | 0 | 0 | 0 | 0 | 0 | 0 | 0 | 1 | 1 | 1 |
| SDEDIT | 0 | 0 | 0 | 1 | 0 | 0 | 0 | 0 | 1 | 0 | 1 | 0 | 0 | 0 | 1 | 0 | 1 | 1 |
| SEAN | 0 | 0 | 0 | 0 | 0 | 1 | 0 | 0 | 0 | 0 | 1 | 0 | 0 | 0 | 1 | 0 | 1 | 0 |
| SEMANTIC | 0 | 1 | 0 | 0 | 0 | 1 | 0 | 0 | 0 | 0 | 1 | 0 | 0 | 0 | 1 | 0 | 0 | 1 |
| SGAN | 1 | 0 | 0 | 0 | 0 | 1 | 0 | 0 | 0 | 0 | 0 | 1 | 0 | 0 | 1 | 0 | 0 | 1 |
| SNGAN | 1 | 0 | 0 | 0 | 0 | 1 | 0 | 0 | 0 | 0 | 1 | 0 | 0 | 0 | 1 | 0 | 1 | 0 |
| SOFT_GAN | 1 | 0 | 0 | 0 | 0 | 1 | 0 | 0 | 0 | 0 | 0 | 1 | 0 | 0 | 1 | 0 | 0 | 1 |
| SRFLOW | 0 | 0 | 1 | 0 | 0 | 0 | 1 | 0 | 0 | 1 | 0 | 0 | 0 | 0 | 0 | 1 | 0 | 0 |
| SRRNET | 1 | 0 | 0 | 0 | 0 | 1 | 0 | 0 | 0 | 0 | 1 | 0 | 0 | 0 | 1 | 0 | 1 | 1 |
| STANDARD_VAE | 0 | 0 | 0 | 0 | 0 | 0 | 0 | 1 | 0 | 1 | 0 | 0 | 0 | 0 | 1 | 0 | 1 | 1 |
| STA_DM_15 | 1 | 0 | 0 | 0 | 0 | 0 | 0 | 1 | 0 | 0 | 1 | 0 | 0 | 0 | 1 | 0 | 1 | 1 |
| STA_DM_21 | 1 | 0 | 0 | 0 | 0 | 0 | 0 | 1 | 0 | 0 | 1 | 0 | 0 | 0 | 1 | 0 | 1 | 1 |
| STA_DM_XL | 1 | 0 | 0 | 0 | 0 | 0 | 0 | 1 | 0 | 0 | 1 | 0 | 0 | 0 | 1 | 0 | 1 | 1 |
| STARGAN | 0 | 1 | 0 | 0 | 0 | 1 | 0 | 0 | 0 | 0 | 1 | 0 | 0 | 0 | 1 | 0 | 0 | 1 |
| STARGAN_2 | 0 | 1 | 0 | 0 | 0 | 0 | 1 | 0 | 0 | 0 | 0 | 1 | 0 | 0 | 1 | 0 | 0 | 1 |
| STGAN | 1 | 0 | 0 | 0 | 0 | 1 | 0 | 0 | 0 | 0 | 0 | 1 | 0 | 0 | 0 | 1 | 1 | 1 |
| STYLEGAN | 0 | 1 | 0 | 0 | 0 | 0 | 1 | 0 | 0 | 0 | 0 | 1 | 0 | 0 | 0 | 1 | 0 | 1 |
| STYLEGAN_2 | 0 | 1 | 0 | 0 | 0 | 0 | 1 | 0 | 0 | 0 | 0 | 1 | 0 | 0 | 0 | 1 | 0 | 1 |
| STYLEGAN_ADA | 0 | 1 | 0 | 0 | 0 | 0 | 1 | 0 | 0 | 0 | 0 | 1 | 0 | 0 | 0 | 1 | 0 | 1 |
| SURVAE_M | 0 | 0 | 1 | 0 | 1 | 0 | 0 | 0 | 0 | 1 | 0 | 0 | 0 | 0 | 1 | 0 | 0 | 0 |
| SURVAE_N | 0 | 0 | 1 | 0 | 1 | 0 | 0 | 0 | 0 | 1 | 0 | 0 | 0 | 0 | 1 | 0 | 0 | 0 |
| TPGAN | 1 | 0 | 0 | 0 | 0 | 0 | 0 | 1 | 0 | 0 | 0 | 1 | 0 | 1 | 0 | 0 | 1 | 1 |
| UGAN | 1 | 0 | 0 | 0 | 0 | 0 | 0 | 1 | 0 | 0 | 1 | 0 | 0 | 0 | 1 | 0 | 1 | 0 |
| UNIT | 0 | 1 | 0 | 0 | 0 | 1 | 0 | 0 | 0 | 0 | 1 | 0 | 0 | 0 | 0 | 1 | 1 | 1 |
| VAE_FIELD | 0 | 0 | 1 | 0 | 0 | 0 | 0 | 1 | 0 | 1 | 0 | 0 | 0 | 0 | 1 | 0 | 0 | 0 |
| VAE_FLOW | 0 | 0 | 1 | 0 | 0 | 0 | 0 | 1 | 0 | 1 | 0 | 0 | 0 | 0 | 1 | 0 | 0 | 0 |
| VAE_GAN | 1 | 0 | 0 | 0 | 0 | 1 | 0 | 0 | 0 | 0 | 1 | 0 | 0 | 0 | 1 | 0 | 0 | 0 |
| VDVAE | 0 | 0 | 1 | 0 | 1 | 0 | 0 | 0 | 0 | 0 | 0 | 1 | 0 | 0 | 0 | 1 | 1 | 1 |
| WGAN | 1 | 0 | 0 | 0 | 0 | 1 | 0 | 0 | 0 | 0 | 1 | 0 | 0 | 0 | 1 | 0 | 0 | 0 |
| WGAN_DRA | 1 | 0 | 0 | 0 | 0 | 1 | 0 | 0 | 0 | 0 | 1 | 0 | 0 | 0 | 1 | 0 | 1 | 0 |
| WGAN_WC | 1 | 0 | 0 | 0 | 0 | 1 | 0 | 0 | 0 | 0 | 1 | 0 | 0 | 0 | 1 | 0 | 1 | 0 |
| WGANGP | 1 | 0 | 0 | 0 | 0 | 1 | 0 | 0 | 0 | 0 | 1 | 0 | 0 | 0 | 1 | 0 | 0 | 0 |
| YLG | 1 | 0 | 0 | 0 | 0 | 1 | 0 | 0 | 0 | 0 | 1 | 0 | 0 | 0 | 0 | 1 | 1 | 1 |

Table 17: Ground truth feature vector used for prediction of Continuous Architecture Hyperparameters for all GMs. The discrete architecture hyperparameter ground truth is defined in (Tab. 6). F1: # layers, F2: # convolutional layers, F3: # fully connected layers, F4: # pooling layers, F5: # normalization layers, F6: #filters, F7: # blocks, F8:# layers per block, and F9: # parameters.

| GM | Layer # | Conv. # | FC # | Pool # | Norm. # | Filter # | Block # | Block Layer # | Para. # |
|---|---|---|---|---|---|---|---|---|---|
| AAE | 9 | 0 | 7 | 0 | 2 | 0 | 0 | 0 | 1, 593, 378 |
| ACGAN | 18 | 10 | 1 | 0 | 7 | 2, 307 | 5 | 3 | 4, 276, 739 |
| ADAGAN_C | 35 | 14 | 13 | 1 | 7 | 4, 131 | 9 | 3 | 9, 416, 196 |
| ADAGAN_P | 35 | 14 | 13 | 1 | 7 | 4, 131 | 9 | 3 | 9, 416, 196 |
| ADM_G_128 | 223 | 90 | 37 | 8 | 88 | N/A | 34 | 7 | 421, 529, 606 |
| ADM_G_256 | 266 | 107 | 45 | 11 | 103 | N/A | 39 | 7 | 553, 838, 086 |
| ADV_FACES | 45 | 23 | 1 | 1 | 20 | 2, 627 | 4 | 6 | 30, 000, 000 |
| ALAE | 33 | 25 | 8 | 0 | 0 | 4, 094 | 3 | 8 | 50, 200, 000 |
| BEGAN | 10 | 9 | 1 | 0 | 0 | 515 | 2 | 4 | 7, 278, 472 |
| BETA_B | 7 | 4 | 3 | 0 | 0 | 99 | 1 | 3 | 469, 173 |
| BETA_H | 7 | 4 | 3 | 0 | 0 | 99 | 1 | 3 | 469, 173 |
| BETA_TCVAE | 7 | 4 | 3 | 0 | 0 | 99 | 1 | 3 | 469, 173 |
| BGAN | 8 | 0 | 5 | 0 | 3 | 0 | 2 | 3 | 1, 757, 412 |
| BICYCLE_GAN | 25 | 14 | 1 | 0 | 10 | 4, 483 | 2 | 10 | 23, 680, 256 |
| BIGGAN_128 | 63 | 21 | 1 | 0 | 41 | 6, 123 | 6 | 10 | 50, 400, 000 |
| BIGGAN_256 | 75 | 25 | 1 | 0 | 49 | 7, 215 | 6 | 12 | 55, 900, 000 |
| BIGGAN_512 | 87 | 29 | 1 | 0 | 57 | 8, 365 | 6 | 14 | 56, 200, 000 |
| Blended_DM | 266 | 107 | 45 | 11 | 103 | N/A | 39 | 7 | 553, 838, 086 |
| CADGAN | 8 | 4 | 1 | 0 | 3 | 451 | 3 | 2 | 3, 812, 355 |
| CCGAN | 22 | 12 | 0 | 0 | 10 | 3, 203 | 2 | 9 | 29, 257, 731 |
| CGAN | 8 | 0 | 5 | 0 | 3 | 0 | 2 | 3 | 1, 757, 412 |
| COCO_GAN | 19 | 9 | 1 | 0 | 9 | 2, 883 | 3 | 4 | 50, 000, 000 |
| COGAN | 9 | 5 | 0 | 0 | 4 | 259 | 2 | 2 | 1, 126, 790 |
| COLOUR_GAN | 19 | 10 | 0 | 0 | 9 | 2, 435 | 2 | 9 | 19, 422, 404 |
| CONT_ENC | 19 | 11 | 0 | 0 | 8 | 5, 987 | 2 | 8 | 40, 401, 187 |
| CONTRAGAN | 35 | 14 | 13 | 1 | 7 | 4, 131 | 9 | 3 | 9, 416, 196 |
| CONTROLNET | 427 | 121 | 132 | 0 | 174 | N/A | 56 | 7 | 39, 726, 979 |
| COUNCIL_GAN | 62 | 30 | 3 | 0 | 29 | 6, 214 | 2 | 10 | 69, 616, 944 |
| CLIPDM | 226 | 120 | 34 | 0 | 72 | N/A | 38 | 3 | 113, 673, 219 |
| CRAMER_GAN | 9 | 4 | 1 | 0 | 4 | 454 | 2 | 3 | 9, 681, 284 |
| CRGAN_C | 35 | 14 | 13 | 1 | 7 | 4, 131 | 9 | 3 | 9, 416, 196 |
| CRGAN_P | 35 | 14 | 13 | 1 | 7 | 4, 131 | 9 | 3 | 9, 416, 196 |
| CYCLEGAN | 47 | 24 | 0 | 0 | 23 | 2, 947 | 4 | 9 | 11, 378, 179 |
| DAGAN_C | 35 | 14 | 13 | 1 | 7 | 4, 131 | 9 | 3 | 9, 416, 196 |
| DAGAN_P | 35 | 14 | 13 | 1 | 7 | 4, 131 | 9 | 3 | 9, 416, 196 |
| DCGAN | 9 | 4 | 1 | 0 | 4 | 454 | 2 | 3 | 9, 681, 284 |
| DDiFFGAN_32 | 289 | 80 | 91 | 0 | 118 | N/A | 40 | 2 | 48, 432, 515 |
| DDPM_32 | 164 | 89 | 24 | 0 | 51 | N/A | 28 | 7 | 35, 746, 307 |
| DDPM_256 | 225 | 120 | 34 | 0 | 71 | N/A | 39 | 7 | 113, 673, 219 |
| DEEPFOOL | 95 | 92 | 1 | 2 | 0 | 7, 236 | 4 | 10 | 22, 000, 000 |
| DFCVAE | 45 | 22 | 2 | 0 | 21 | 4, 227 | 4 | 7 | 2, 546, 234 |
| DIFF_ISGEN | 88 | 24 | 56 | 0 | 8 | N/A | 8 | 6 | 30, 276, 583 |
| DIFF_PGAN | 45 | 20 | 0 | 3 | 13 | N/A | 11 | 4 | 105, 684, 175 |
| DIFF_SGAN | 48 | 20 | 28 | 0 | 8 | N/A | 8 | 6 | 24, 767, 458 |
| DIFFAE | 712 | 263 | 171 | 45 | 233 | N/A | 118 | 7 | 336, 984, 582 |
| DIFFAE_LATENT | 717 | 264 | 172 | 46 | 235 | N/A | 155 | 6 | 445, 203, 974 |
| DISCOGAN | 21 | 12 | 0 | 0 | 9 | 3, 459 | 2 | 9 | 29, 241, 731 |
| DRGAN | 44 | 28 | 1 | 1 | 14 | 4, 481 | 3 | 8 | 18, 885, 068 |
| DRIT | 19 | 10 | 0 | 0 | 9 | 1, 793 | 4 | 3 | 9, 564, 170 |
| DUALGAN | 25 | 14 | 1 | 0 | 10 | 4, 483 | 2 | 10 | 23, 680, 256 |
| EBGAN | 6 | 3 | 1 | 0 | 2 | 195 | 2 | 2 | 738, 433 |
| ESRGAN | 66 | 66 | 0 | 0 | 0 | 4, 547 | 5 | 4 | 7, 012, 163 |
| FACTOR_VAE | 7 | 4 | 3 | 0 | 0 | 99 | 1 | 3 | 469, 173 |
| Fast pixel | 17 | 9 | 0 | 0 | 8 | 768 | 2 | 8 | 4, 600, 000 |
| FFGAN | 39 | 19 | 1 | 1 | 19 | 3, 261 | 0 | 0 | 50, 000, 000 |
| FGAN | 5 | 0 | 3 | 0 | 2 | 0 | 2 | 2 | 2, 256, 401 |
| FGAN_KL | 5 | 0 | 3 | 0 | 2 | 0 | 2 | 2 | 2, 256, 401 |
| FGAN_NEYMAN | 5 | 0 | 3 | 0 | 2 | 0 | 2 | 2 | 2, 256, 401 |
| FGAN_PEARSON | 5 | 0 | 3 | 0 | 2 | 0 | 2 | 2 | 2, 256, 401 |
| FGSM | 95 | 92 | 1 | 2 | 0 | 7, 236 | 4 | 10 | 22, 000, 000 |
| FPGAN | 23 | 12 | 0 | 0 | 11 | 2, 179 | 2 | 6 | 53, 192, 576 |
| FSGAN | 37 | 20 | 0 | 1 | 16 | 2, 863 | 4 | 8 | 94, 669, 184 |
| FVBN | 28 | 0 | 28 | 0 | 0 | 0 | 1 | 1 | 307, 721 |
| GAN_ANIME | 25 | 18 | 0 | 0 | 7 | 2, 179 | 4 | 6 | 8, 467, 854 |
| Gated_pixel_cnn | 32 | 32 | 0 | 0 | 0 | 5, 433 | 3 | 10 | 3, 364, 161 |
| GDWCT | 79 | 27 | 40 | 1 | 11 | 5, 699 | 2 | 4 | 51, 965, 832 |
| GFLM | 95 | 92 | 1 | 2 | 0 | 7, 236 | 4 | 10 | 22, 000, 000 |
| GGAN | 8 | 4 | 1 | 0 | 3 | 451 | 3 | 2 | 3, 812, 355 |
| GLIDE | 331 | 93 | 103 | 6 | 129 | N/A | 74 | 5 | 385, 030, 726 |
| ICRGAN_C | 35 | 14 | 13 | 1 | 7 | 4, 131 | 9 | 3 | 9, 416, 196 |
| ICRGAN_P | 35 | 14 | 13 | 1 | 7 | 4, 131 | 9 | 3 | 9, 416, 196 |

Table 18: Ground truth feature vector used for prediction of Continuous Architecture Hyperparameters for all GMs. The discrete architecture hyperparameter ground truth is defined in (Tab. 6). F1: # layers, F2: # convolutional layers, F3: # fully connected layers, F4: # pooling layers, F5: # normalization layers, F6: #filters, F7: # blocks, F8:# layers per block, and F9: # parameters.

| GM | Layer # | Conv. # | FC # | Pool # | Norm. # | Filter # | Block # | Block Layer # | Para. # |
|---|---|---|---|---|---|---|---|---|---|
| IDDPM_32 | 193 | 85 | 32 | 0 | 76 | N/A | 45 | 2 | 52, 546, 438 |
| IDDPM_64 | 195 | 87 | 32 | 0 | 76 | N/A | 45 | 2 | 27, 3049, 350 |
| IDDPM_256 | 201 | 96 | 34 | 0 | 71 | N/A | 40 | 5 | 113, 676, 678 |
| ILVER_256 | 266 | 107 | 45 | 11 | 103 | N/A | 39 | 7 | 553, 838, 086 |
| Image_GPT | 59 | 42 | 0 | 0 | 17 | 4, 673 | 7 | 8 | 401, 489 |
| INFOGAN | 7 | 3 | 1 | 0 | 3 | 195 | 2 | 2 | 1, 049, 985 |
| LAPGAN | 11 | 6 | 5 | 0 | 0 | 262 | 4 | 2 | 2, 182, 857 |
| LDM | 255 | 127 | 24 | 0 | 104 | N/A | 65 | 4 | 329, 378, 945 |
| LDM_CON | 503 | 159 | 184 | 8 | 152 | N/A | 65 | 8 | 456, 755, 873 |
| Lmconv | 105 | 60 | 10 | 35 | 0 | 7, 156 | 15 | 5 | 46, 000, 000 |
| LOGAN | 35 | 14 | 13 | 1 | 7 | 4, 131 | 9 | 3 | 9, 416, 196 |
| LSGAN | 9 | 5 | 0 | 0 | 4 | 1, 923 | 2 | 4 | 23, 909, 265 |
| MADE | 2 | 0 | 2 | 0 | 0 | 0 | 1 | 2 | 12552784 |
| MAGAN | 9 | 5 | 0 | 0 | 4 | 963 | 2 | 3 | 11, 140, 934 |
| MEMGAN | 14 | 7 | 1 | 0 | 6 | 1, 155 | 3 | 4 | 4, 128, 515 |
| MMD_GAN | 9 | 4 | 1 | 0 | 4 | 454 | 2 | 3 | 9, 681, 284 |
| MRGAN | 9 | 4 | 1 | 0 | 4 | 451 | 3 | 2 | 15, 038, 350 |
| MSG_STYLE_GAN | 33 | 25 | 8 | 0 | 0 | 4, 094 | 3 | 8 | 50, 200, 000 |
| MUNIT | 18 | 15 | 0 | 0 | 3 | 3, 715 | 2 | 6 | 10, 305, 035 |
| NADE | 1 | 0 | 1 | 0 | 0 | 0 | 1 | 1 | 785, 284 |
| OCFGAN | 9 | 4 | 1 | 0 | 4 | 454 | 2 | 3 | 9, 681, 284 |
| PGD | 95 | 92 | 1 | 2 | 0 | 7, 236 | 4 | 10 | 22, 000, 000 |
| PIX2PIX | 29 | 16 | 0 | 0 | 13 | 5, 507 | 2 | 13 | 54, 404, 099 |
| PixelCNN | 17 | 9 | 0 | 0 | 8 | 768 | 2 | 8 | 4, 600, 000 |
| PixelCNN++ | 105 | 60 | 10 | 35 | 0 | 7, 156 | 15 | 5 | 46, 000, 000 |
| PIXELDA | 27 | 14 | 1 | 0 | 12 | 835 | 4 | 6 | 483, 715 |
| PixelSnail | 90 | 90 | 0 | 0 | 0 | 4, 051 | 3 | 10 | 40, 000, 000 |
| PNDM_32 | 164 | 89 | 24 | 0 | 51 | N/A | 28 | 7 | 35, 746, 307 |
| PNDM_256 | 266 | 107 | 45 | 11 | 103 | N/A | 39 | 7 | 553, 838, 086 |
| PROG_GAN | 26 | 25 | 1 | 0 | 0 | 4, 600 | 3 | 8 | 46, 200, 000 |
| RGAN | 7 | 3 | 1 | 0 | 3 | 195 | 2 | 2 | 1, 049, 985 |
| RSGAN_HALF | 8 | 4 | 1 | 0 | 3 | 899 | 3 | 2 | 13, 129, 731 |
| RSGAN_QUAR | 8 | 4 | 1 | 0 | 3 | 451 | 3 | 2 | 3, 812, 355 |
| RSGAN_REG | 8 | 4 | 1 | 0 | 3 | 1, 795 | 3 | 2 | 48, 279, 555 |
| RSGAN_RES_BOT | 15 | 7 | 1 | 0 | 7 | 963 | 3 | 4 | 758, 467 |
| RSGAN_RES_HALF | 15 | 7 | 1 | 0 | 7 | 1, 155 | 3 | 4 | 1, 201, 411 |
| RSGAN_RES_QUAR | 15 | 7 | 1 | 0 | 7 | 579 | 3 | 4 | 367, 235 |
| RSGAN_RES_REG | 15 | 7 | 1 | 0 | 7 | 2, 307 | 3 | 4 | 4, 270, 595 |
| SAGAN | 11 | 6 | 1 | 0 | 4 | 139 | 2 | 4 | 16, 665, 286 |
| SEAN | 19 | 16 | 0 | 0 | 0 | 5, 062 | 2 | 7 | 266, 907, 367 |
| SEMANTIC | 23 | 12 | 0 | 0 | 11 | 2, 179 | 2 | 6 | 53, 192, 576 |
| SGAN | 7 | 3 | 1 | 0 | 3 | 195 | 2 | 2 | 1, 049, 985 |
| SCOREDIFF_256 225 | 120 | 34 | 0 | 11 | 71 | N/A | 39 | 7 | 113, 673, 219 |
| SDEdit | 226 | 120 | 34 | 0 | 72 | N/A | 38 | 3 | 113, 673, 219 |
| SNGAN | 23 | 11 | 1 | 0 | 11 | 3, 871 | 4 | 5 | 10, 000, 000 |
| SOFT_GAN | 8 | 0 | 5 | 0 | 3 | 0 | 2 | 3 | 1, 757, 412 |
| SRFLOW | 66 | 66 | 0 | 0 | 2 | 4, 547 | 5 | 4 | 7, 012, 163 |
| SRRNET | 74 | 36 | 1 | 0 | 37 | 2, 819 | 4 | 16 | 4, 069, 955 |
| STANDARD_VAE | 7 | 4 | 3 | 0 | 0 | 99 | 1 | 3 | 469, 173 |
| STARGAN | 23 | 12 | 0 | 0 | 11 | 2, 179 | 2 | 6 | 53, 192, 576 |
| STARGAN_2 | 67 | 26 | 12 | 4 | 25 | 4, 188 | 4 | 12 | 94, 008, 488 |
| STA_DM_15 | 503 | 159 | 184 | 8 | 152 | N/A | 65 | 8 | 456, 755, 873 |
| STA_DM_21 | 503 | 159 | 184 | 8 | 152 | N/A | 65 | 8 | 456, 755, 873 |
| STA_DM_XL | 601 | 184 | 201 | 12 | 185 | N/A | 74 | 9 | 618, 997, 638 |
| STGAN | 19 | 10 | 0 | 0 | 9 | 2, 953 | 2 | 5 | 25, 000, 000 |
| STYLEGAN | 33 | 25 | 8 | 0 | 0 | 4, 094 | 3 | 8 | 50, 200, 000 |
| STYLEGAN_2 | 33 | 25 | 8 | 0 | 0 | 4, 094 | 3 | 8 | 59, 000, 000 |
| STYLEGAN2_ADA | 33 | 25 | 8 | 0 | 0 | 4, 094 | 3 | 8 | 59, 000, 000 |
| SURVAE_FLOW_MAX | 95 | 90 | 0 | 5 | 0 | 6, 542 | 2 | 20 | 25, 000, 000 |
| SURVAE_FLOW_NON | 90 | 90 | 0 | 0 | 0 | 6, 542 | 2 | 20 | 25, 000, 000 |
| TPGAN | 45 | 31 | 2 | 1 | 11 | 5, 275 | 0 | 0 | 27, 233, 200 |
| UGAN | 9 | 4 | 1 | 0 | 4 | 771 | 2 | 3 | 4, 850, 692 |
| UNIT | 43 | 22 | 0 | 0 | 21 | 4, 739 | 4 | 8 | 13, 131, 779 |
| VAE_field | 6 | 0 | 6 | 0 | 0 | 0 | 1 | 3 | 300, 304 |
| VAE_flow | 14 | 0 | 14 | 0 | 0 | 0 | 2 | 4 | 760, 448 |
| VAEGAN | 17 | 7 | 2 | 0 | 8 | 867 | 2 | 6 | 26, 396, 740 |
| VDVAE | 48 | 42 | 0 | 6 | 0 | 3, 502 | 3 | 13 | 41, 000, 000 |
| WGAN | 9 | 5 | 0 | 0 | 4 | 1, 923 | 2 | 4 | 23, 909, 265 |
| WGAN_DRA | 18 | 10 | 1 | 0 | 7 | 2, 307 | 5 | 3 | 4, 276, 739 |
| WGAN_WC | 18 | 10 | 1 | 0 | 7 | 2, 307 | 5 | 3 | 4, 276, 739 |
| WGANGP | 9 | 5 | 0 | 0 | 4 | 1, 923 | 2 | 4 | 23, 905, 841 |
| YLG | 33 | 20 | 1 | 2 | 10 | 5, 155 | 5 | 5 | 42, 078, 852 |

