# OpenReview forum: "Tracing Hyperparameter Dependencies for Model Parsing via Learnable Graph Pooling Network"
_NeurIPS.cc/2024/Conference — NeurIPS 2024 poster_

### Official Review · Reviewer_Q98w · 2024-07-11

**Soundness:** 3
**Presentation:** 3
**Contribution:** 3
**Rating:** 5
**Confidence:** 3

**Summary:**

This paper studies the model parsing (MP) task [1], which predicts the hyperparameters of the generative model with a generated image input. The authors propose a Learnable Graph Pooling Network (LGPN) method to explore hyperparameter dependencies in the generative model. The LGPN consists of two core designs: the Generation Trace Capturing Network (GTC) and the learnable pooling-unpooling graph mechanism called the GCN Refinement Block. Extensive experiments validate the effectiveness of the method.

**Strengths:**

1. A new idea of graph pooling mechanisms to model hyperparameter dependency in the MP task.
2. A new dataset, RED140, extended from RED116 [1].
3. Outstanding performances on both the RED116 and RED140 datasets.

**Weaknesses:**

Despite the strengths mentioned above, there are some concerns listed as follows:
1. The authors provide a new dataset, RED140, which removes and collects some samples. Please clarify the distinction between these two datasets. Is the ground-truth feature vector for Discrete Architecture hyperparameters newly added to supervise the model training?
2. The work mainly compares with FEN-PN. [1]. The experimental results on the RED116 dataset in Fig. 5(c) just report the F1 score and L1 error performance. [1] provides more evaluation metrics (P-value, Corr. Coef, Slope, Acc., etc.) and tests more dataset types and GM models, such as continuous type, discrete type, and loss type datasets, as well as face and non-face GMs. These analyses provide more evidence to show the method's generalizability and robustness. By comparison, the proposed method needs more experimental evidence.
3. From Fig. 6 (b), the performance of the proposed method has a slight drop compared with FEN-PN [1]. What happens?
4. In lines 30-32, there is a motivation statement for hyperparameter dependency modeling: ”For instance, an inherent dependency exists between GM’s layer number and parameter number.” The authors do not provide such analysis and discussion in the experiments. The discussion on hyperparameter dependency analyses (what and how dependents?) is insufficient.
5. About the correlation graph construction (lines 125–126), “we count the occurrence of such pairs in the RED140 to retrieve the matrix G\in R^{C \times C}, where C is the hyperparameter number.” The method approach appears to overuse the dataset information in advance and is unfair when compared to FEN-PN [1].
6. There are some parameters, such as the thresholds to remove relations in the graph or binary classification; how about the sensitivities of these parameters?

**Questions:**

Please see the comments in Weakness. I'm willing to change my rating if the authors address my concerns.

**Limitations:**

The authors introduce the performance improvement, but the discussions on hyperparameter dependency analyses (what and how dependents?) and the compatibility on different GM models are insufficient.

---

> ### Author Rebuttal · Authors · 2024-08-06
>
> We thank the reviewer for constructive feedbacks. Reviews are positive about our empirical performance and the learnable pooling algorithm, and we address all concerns raised as follows:
>
> **Brief Recap:**
>
> We adhere to the problem statement of model parsing [R3], in which the algorithm takes one image as the input and predicts 37 hyperparameters used in the generative model (GM) that generates the input image.
> These 37 predictable hyperparameters are clearly defined in [R3] and detailed in our supplementary, including discrete architecture hyperparameters, continuous architecture hyperparameters, and loss functions.
>
> **Q1: The difference between RED116 dataset and RED140 dataset & Ground truth vector:**
>
> RED116 is a subset of RED140, with RED140 serving as a more advanced model parsing dataset. The key distinctions between RED116 and RED140 are as follows:
>
> 1. RED116 does not have diffusion model images, whereas RED140 includes images from 24 distinct diffusion models (details in supplementary Sec.D and E). Consequently, RED140 provides a richer variety of samples, enhancing the dataset's utility in developing algorithms capable of identifying hyperparameter patterns in images generated by diffusion models.
>
> 2. RED140 has real images covering different semantics (supplementary Sec.D details). These real images are crucial for training models to only predict hyperparameters for generated images, rather than real ones. Without these real images, models might erroneously assign hyperparameters to real images when used as inputs
>
> Moreover, each GM has an annotated ground truth vector for discrete architecture hyperparameters.  As defined in [R3], these vectors can supervise the training. FEN-PN and our LGPN use the same train and test protocol.
>
> **Q2: The comparison on RED116 & comparison on face and non-face GM:**
>
> Thank you very much for the suggestion.
> Please refer to the general rebuttal response, in which we use Table R.1, R.2, R.3, and R.4 to report various performances on RED116.
> Also, the main paper's Table 1 shows the performance of RED140.
> These five tables show that LGPN surpasses [R3] on all metrics (i.e., discrete architecture hyperparameters, continuous architecture hyperparameters, and loss functions) on both RED116 and RED140 datasets.
> Such performance superiority indicates our proposed method's better generalization ability and robustness.
>
> **Q3: Image attribution performance:**
>
> The main paper's Fig.6 (b) shows a relatively easy protocol in which different models have similar saturated performances. Therefore, we extend this image attribution experiment into Table R.9 with a more challenging protocol, showcasing our method's performance superiority to FEN-PN[R3]:
>
> - **Protocol 1**: as Fig.6 (b) reports (detailed in Supplementary Fig.8), the image attribution is a 5-way classification task --- classifying if the image is real or generated by which one from 4 GMs (e.g., SNGAN, MMDGAN, CRAMERGAN, and ProGAN).
>
> - **Protocol 2**: On top of protocol 1, we add 2 more generative methods, resulting in a more challenging task: a 7-way classification task, classifying whether the image is real samples or generated by which one from 6 GMs (e.g., styleGANv2, styleGANv3, SNGAN, MMDGAN, CRAMERGAN, and ProGAN).
>
> **Table R.9** Image attribution performance.
>
> |Method|protocol 1|protocol 2|
> |:----:|:--------:|:--------:|
> ||CelebA/LSUN|CelebA/LSUN|
> |PRNU|86.61/67.84|74.23/67.92|
> |Attr.|99.43/98.58|84.16/81.63|
> |FEN-PN|99.66/**99.84**|81.09/78.28|
> |Ours|**99.79**/99.73|**88.11**/**86.23**|
>
> **Q4: Hyperparameter dependencies:**
>
> Sorry for the inconvenience. One common dependency is that GMs with more layers usually have more parameters (lines 30-32). This is illustrated by lines 270-274 and Tab. 2(b):
>
> - Lines 270-274 detail that, with capturing the dependency between _Param. Num._ and _Layer Num._,  the GTC with GCN refinement block achieves a better performance than GTC with MLP layers, on predicting these two continuous hyperparameters.
>
> - Tab. 2(b) shows that the GCN refinement block helps GTC achieve better prediction performance than GTC with MLP layers on all continuous architecture hyperparameters. Such performance improvements are attributed to the proposed GCN refinement block, which effectively captures dependencies among continuous hyperparameters, such as FC layer numbers, model parameter numbers, pooling layer numbers, etc.
>
> **Q5: Graph construction:**
>
> Sorry for the confusion. We only use training samples from RED140 dataset to construct such a graph and strictly forbid data leakage. The procedure of using training samples for graph construction is commonly adopted in prior works [10, 7, 59, 39, 15, 50].
> In the revised version, we will add the statement --- "use training samples from the RED140 dataset."
>
> **Q6: Sensitivity analysis:**
>
> Thanks for this valid concern.
> We follow your suggestions and show the model parsing performance with different thresholds in Table R.10.
> Overall, the best performance is achieved when the threshold is 0.45, as we employ in LGPN.
> Also, this best performer is comparable to models that use 0.35 and 0.55 as threshold values, showing LGPN's robustness to different thresholds.
> However, the performance decreases more when the threshold increases to 0.65, which causes the correlation graph to have very sparse connectivities, hindering the learning of hyperparameter dependencies.
>
> The second robustness analysis is Table 7 in the supplementary, which shows that our model parsing performance is generally robust to using different numbers of graph nodes to represent each continuous hyperparameter.
>
> **Table R.10** The model performance on different graph thresholds.
>
> ||Loss Func.|Dis. Archi. Para.|Con. Archi. Para.|
> |:---:|:----------:|:----------:|:-----:|
> |Threshold|F1/Acc|F1/Acc|L1 error|
> |0.35|84.0/83.0|79.2/77.0|0.122|
> |0.45|**84.6**/**83.3**|**79.5**/**77.5**|**0.120**|
> |0.55|84.5/82.8|78.9/77.0|0.124|
> |0.65|82.7/82.5|77.0/74.5|0.139|

---

> > ### Author Response · Authors · 2024-08-12
> > **Thanks to Reviewer Q98w**
> >
> > Dear Reviewer Q98w:
> >
> > Please allow us to sincerely thank you again for reviewing our paper and the valuable feedback, and in particular for recognizing the strengths of our paper in terms of the reasonable algorithm design and high effectiveness.
> >
> > Please let us know if our response and additional experiments have properly addressed your concerns about the comparative experiment and motivations. We are more than happy to answer any additional questions during the discussion period. Your feedback will be greatly appreciated!
> >
> > Best,
> >
> > Paper14633 Authors

---

> > ### Comment · Reviewer_Q98w · 2024-08-12
> >
> > Thank the authors for the response. Tracking hyperparameter dependencies is the core topic of this work, but the reply to the discussion is still insufficient. This work appears to focus on "intergrating hyperparameters" rather than "tracing" the hyperparameter dependencies. The training and inference costs compared to the others should be discussed.

---

> > > ### Author Response · Authors · 2024-08-13
> > > **The Rebuttal Response to Reviewer Q98w**
> > >
> > > We sincerely thank Reviewer Q98w again for valuable suggestions, and these are our precise answers after diligently examining questions and comments.
> > >
> > > >**Q7. "what and how to depend" examples**
> > >
> > > Thank you for this insightful question. **Q5** from the first rebuttal response shows that Tab. 2(b) and Lines 270-274 illustrate the idea of hyperparameter dependency claimed in lines 30-32. Furthermore, these are examples of "what and how to depend":
> > >
> > > 1. There exists a dependency between _Layer Num._ and _Parameter Num._. In other words, in general, Layer Num. **positively depends** on Parameter Num., and vice versa.
> > > This is because generative models, which have a large Layer Num., very likely also have a large Parameter Num.
> > > Such a dependency (i.e., correlation) is helpful in model parsing, as when our method confidently predicts that the generative model has a large Layer Num., then this generative model can likely have a large Parameter Num.
> > > As a result, LGPN's GCN refinement block captures this dependency and achieves **0.147** and **0.081 L1 error** on predicting Layer Num. and Param. Num., whereas GTC with MLP layers can only have **0.149** and **0.148 L1 error**, respectively.
> > >
> > > 2. There exists a dependency between _Conv. Layer Num._ and _Conv. Filters Num_. (i.e., _Filters Num_ in the Tab. 2(b)).
> > > Apparently, Conv. Layer Num. **positively depends** on Filters Num., and vice versa. For example, generative models with more convolution layers, in theory, have a proportionally large number of convolution filters.
> > > This dependency can be used in model parsing: when our method has the confident prediction that the generative model has a small number of Conv. Layer Num., then it is likely that this generative model also has a small number of Filters Num.
> > > Such a dependency is captured by the GCN refinement block, which achieves **0.137 and 0.149 L1 error** on predicting Conv. Layer Num. and Filters Num., whereas GTC with MLP layers can only have **0.151 and 0.161 L1 error**, respectively. Performance detailed in Tab.2(b).
> > >
> > > 3. There is a dependency between L1 and L2 loss functions: the existence of L1 loss function **negatively depends** on the existence of L2 loss function.
> > > It's important to note that the generative model typically doesn't use L1 and L2 loss functions simultaneously, and this correlation (e.g., dependency) is useful in the model parsing task.
> > > Specifically, when our method confidently predicts that the L1 loss function is used by the generative model while finding it difficult to predict if the L2 loss function is used, the model can use the confidence prediction on the L1 loss function as the prior knowledge to decide that L2 loss function is likely not used by the given generative model.
> > > As a result, GTC with GCN refinement block achieves **0.101 and 0.163 higher F1 scores** than GTC with MLP layers on predicting L1 and L2 loss functions, respectively.
> > >
> > > >**Q8. Focus on tracing hyperparameter dependencies**
> > >
> > > As claimed in line 8 and line 34, we use graph nodes to represent hyperparameters, and each graph edge between graph nodes is used to **trace** (or capture) corresponding hyperparameter dependencies.
> > >
> > > One important focus of our work is to learn a graph structure (i.e., $\mathbf{A}^{\prime}_{l}$) with graph edges that **trace hyperparameter dependencies effectively**, helping the representation learning for graph nodes that are used for predicting hyperparameters.
> > > This is demonstrated by our major technical contribution (Sec.3.2.2) --- the GCN refinement block with the graph pooling-unpoling mechanism, which optimizes a trainable graph (e.g., both graph edges and graph nodes are trainable) for better model parsing performance and generalization ability.
> > > Empirically, as detailed in Fig. 5(a), each generative model has its own specific learned graph structure.
> > >
> > > >**Q9. Training and inference costs**
> > >
> > > Thank you for this additional interesting question. Aside from the **Q4** from the response to **Reviewer nGEg**, we use the following Table R.11 to show the inference and training costs compared to FEN-PN [R3] and model variant of GTC with the stacked GCN as the model parsing head.
> > > Specifically, the inference speed is measured by the time cost by a one-forward propagation in the single NVIDIA RTX A6000 GPU with 48G memory.
> > > As shown by Table R.11, our LGPN is a computationally efficient algorithm.
> > > This is because our innovative graph pooling design actually largely reduces the graph size via the learned graph pooling mechanism, offering a largely more computationally efficient than prior GCN-based methods [10,7,59].
> > >
> > > **Table R.11** Different model parsing methods' inference and training cost.
> > >
> > > |Method|Backbone|Model Parsing Head|GFLOPs(↓)|Inference Speed(↓)|Training Time(↓)|
> > > |-------|--------|-------|------|---------------|-------------|
> > > |FEN-PN|FEN|PN|7.1|101.1 ms|around 1 day|
> > > |baseline|GTC|Stacked GCN|4.8|57.5 ms|around 1 day|
> > > |LGPN|GTC|GCN refinement|4.2|41.5 ms|around 1 day|

---

### Official Review · Reviewer_6bSd · 2024-07-15

**Soundness:** 3
**Presentation:** 3
**Contribution:** 2
**Rating:** 5
**Confidence:** 3

**Summary:**

This paper examines the challenge of predicting the hyperparameters of a generative model from its output, specifically a generated image. The proposed approach introduces a learnable graph pooling technique, framing the problem as a node classification task. Experiments across several tasks demonstrate the effectiveness of this method.

**Strengths:**

- This paper explores the interesting problem of predicting hyperparameters and their dependencies.

- The proposed graph-based method is reasonable and appears to be easy to implement.

- The experimental results in Table 1 clearly demonstrate the effectiveness of the proposed method for model parsing. Additionally, ablation studies or case studies provide further valuable insights.

**Weaknesses:**

-  It remains unclear how the proposed method aids in the detection of coordinated attacks.

- Additional demonstrations or examples of predicted hyperparameters would provide clarity on the effectiveness of the proposed method.

**Questions:**

See weakness.

**Limitations:**

The authors have adequately addressed the limitations.

---

> ### Author Rebuttal · Authors · 2024-08-06
>
> Thank the reviewer for recognizing our interesting research topic and easy-to-implement algorithm. In this response, we present clarification of the coordinated attack detection (**Q1**) and additional empirical results (**Q2**) that will be added in the revised version.
>
> **Q1: Coordinated attack detection clarification:**
>
> We follow the procedure defined in the previous work [R3] for the coordinated attacks detection task, which calculates the cosine distance between predicted hyperparameter vectors from two input images (i.e., ${\mathbf{I}}_1$ and ${\mathbf{I}}_2$). If this distance is larger than a certain threshold, we decide two images are from different generative models (GMs); otherwise, they are from the same GM.
>
> More formally, as stated in Sec.3.1 of the main paper, given an image ${\mathbf{I}}$ as the input, predicted outputs from the LGPN are three vectors, e.g., ${\mathbf{y}}^d\in\mathbb{R}^{18}$, ${\mathbf{y}}^c\in\mathbb{R}^{9}$ and ${\mathbf{y}}^l\in\mathbb{R}^{10}$, which represent the predicted discrete architecture hyperparameters, continuous architecture hyperparameters, and loss function, respectively. Then, the final hyperparameter vector (i.e., descriptor) can be a vector ${\mathbf{y}}^t\in\mathbb{R}^{37}$ via concatenating ${\mathbf{y}}^d$, ${\mathbf{y}}^c$, and ${\mathbf{y}}^l$.
>
> Based on this, given two images (i.e., ${\mathbf{I}}_1$ and ${\mathbf{I}}_2$), we use LGPN to retrieve ${\mathbf{y}}^{t1}\in\mathbb{R}^{37}$ and ${\mathbf{y}}^{t2}\in\mathbb{R}^{37}$, respectively. Then, we compute the cosine distance between ${\mathbf{y}}^{t1}\in\mathbb{R}^{37}$ and ${\mathbf{y}}^{t2}\in\mathbb{R}^{37}$. If this distance is larger than a certain threshold, we decide $\mathbf{I}_1$ and $\mathbf{I}_2$ are generated from different generative models (GMs); otherwise, they are from the same GM. This procedure is used by both FEN-PN [R3] and LGPN, and the revised version will include more details.
>
> **Q2: Additional model parsing evidence:**
>
> We agree that more experimental evidence would solidify our contributions. Due to the paper length, we keep a large number of experimental results in the supplementary. Please refer to our supplementary, which already offers additional quantitative results and visualizations as follows:
>
> - Supplementary Figure 7: Prediction performance for each hyperparameter.
> - Supplementary Figure 12: Visualizations of learned correlation matrix similarity.
> - Supplementary Table 9: The detailed performance of RED116.
> - Supplementary Figure 10: Visualization of model parsing performance on different semantics.
> - Supplementary Figure 12: Model parsing performance for different generative models.
> - Supplementary Section D: details of the RED140 dataset.
> - Supplementary Section E: details of the annotated ground truth.
>
> Moreover, we will add our rebuttal responses in the revised version, which includes:
>
> - The performance on RED116 using different metrics. (Table R.1, R.2, R.3)
> - The detailed performance of face and non-face images. (Table R.4)
> - The performance when the model takes a different number of images as the input. (Table R.5)
> - The performance using GCN refinement with different backbones. (Table R.7)
> - The performance with mistaken labeling. (Table R.8)
> - The updated image attribution performance. (Table R.9)
> - The sensitivity of using different thresholds for the correlation graph. (Table R.10)

---

> > ### Author Response · Authors · 2024-08-12
> > **Thanks to Reviewer 6bSd**
> >
> > Dear Reviewer 6bSd:
> >
> > Please allow us to sincerely thank you again for reviewing our paper and the valuable feedback, and in particular for recognizing the strengths of our paper in terms of our interesting research topic and the efficient algorithm.
> >
> > Please let us know if our response and additional experiments have properly addressed your concerns. We are more than happy to answer any additional questions during the discussion period. Your feedback will be greatly appreciated!
> >
> > Best,
> >
> > Paper14633 Authors

---

### Official Review · Reviewer_born · 2024-07-25

**Soundness:** 3
**Presentation:** 3
**Contribution:** 3
**Rating:** 6
**Confidence:** 2

**Summary:**

This paper focuses on the problem of model parsing, which aims to analyze the hyperparameters of the model that generates a given image. Different from existing model parsing methods, the proposed Learnable Graph Pooling Network (LGPN) exploits the dependencies among hyperparameters to better perform model parsing. In LGPN, the features are extracted by Generation Trace Capturing Network (GTC), which fuses the outputs of the high-res branch and the ResNet branch to preserve high-frequency generation artifacts. The performance of LGPN is compared with baseline methods and a recently proposed model parsing method on the RED140 dataset. The effectiveness of the proposed method is verified by the experimental results.

**Strengths:**

1. The considered problem setting is novel and practical.
2. The dependences among hyperparameters are exploited to improve the model parsing performance.
3. The proposed method outperforms the compared methods.

**Weaknesses:**

1. The effectiveness of GTC can be further verified if different backbones are also compared with GCN refinement as the MP head.
2. It is unclear whether the ground truth of the hyperparameter hierarchy assignment is constructed based on the entire dataset or just the training data.
3. The quality of the paper can be improved if the expressions are polished more carefully, especially in the experimental part.

**Questions:**

1. Is the ground truth of the hyperparameter hierarchy assignment constructed based on the entire dataset or just the training data?
2. It has been shown that the hyperparameter hierarchy prediction loss and artifacts isolation loss is important to the performance. Will the performance of the proposed method drop noticeably if: 1) the hyperparameter hierarchy in the test set differ greatly from the training set; 2) some generated images are mistakenly treated as real images in the training set.

**Limitations:**

The limitations are discussed in the conclusion part. There is no potential negative societal impact.

---

> ### Author Rebuttal · Authors · 2024-08-06
>
> We thank the reviewer for acknowledging our practical and novel research topic and the effectiveness of the proposed LGPN. In this response, we offer additional experimental results regarding GTC's effectiveness (**Q1**), a discussion of mistaken labeling (**Q2**), and clarification on the hyperparameter hierarchy assignment (**Q3**).
>
> **Q1: Using GCN refinement as model parsing (MP) head:**
>
> Thanks! Table.R.7 provides additional experimental results to demonstrate the effectiveness of the proposed GTC.
>
> **Table R.7** Performance of different backbones with the GCN refinement block as MP head.
>
> | Backbone | MP head        | Loss function | Dis. Archi. Para. | Con. Archi. Para. |
> |:--------:|:--------------:|:-------------:|:-----------------:|:-----------------:|
> | | |F1/Acc (↑)          |F1/Acc (↑)          |L1 error (↓)        |
> | | |--------------------|--------------------|--------------------|
> | ResNet   | GCN ref. | 80.5 / 82.5   | 75.6 / 69.8       | 0.155             |
> | HR-Net   | GCN ref. | 81.8 / 82.1   | 74.8 / 72.9       | 0.147             |
> | ViT      | GCN ref. | 76.9 / 76.7   | 69.7 / 67.0       | 0.170             |
> | GTC      | GCN ref. | **84.6** / **83.3**   | **79.5** / **77.5** | **0.120**|
>
> **Q2: will the performance drop when some generated images are mistakenly treated as real images in the training set:**
>
> This is an interesting advice! We take this mistaken labeling idea and intentionally treat different percentages of generated images as real ones in the training set. We report the performance in Table R.8.
> Specifically, the model performance is still acceptable when 1% or 5% percent of samples are mistakenly labeled.
> However, as expected, the performance largely decreases when the LGPN experiences 8% mistaken labeling. This is reasonable as the proposed LGPN identifies the generation trace as the foundation for predicting hyperparameters.
> Consequently, this mistaken labeling disturbs the learning of the generation trace, largely reducing the overall prediction performance.
>
> **Table R.8** Performance metrics for different percentages of mistaken labeling.
>
> |Percent| Loss Fun. | Dis. Archi. Para.  |
> |:-:|:------------------:|:------------------:|
> | |F1/Acc(↑)              |F1/Acc(↑)              |
> | |--------------------|--------------------|
> | 0| **84.6** / **83.3** | **79.5** / **77.5** |
> | 1| 83.0 / 80.4       | 77.1 / 73.2        |
> | 5| 80.4 / 79.2       | 74.9 / 70.6        |
> | 8| 74.4 / 61.2       | 68.3 / 45.9        |
>
> **Q3: Clarification on hyperparameter hierarchy assignment:**
>
> Sorry for the confusion. The hyperparameter hierarchy assignment (the main paper's Figure 4) is a fixed constraint, which uses general knowledge to encourage the model to pool graph nodes that represent hyperparameters with the same attribute. For example, graph nodes for pixel-wise loss functions (e.g., L1 and L2) are merged, and graph nodes for normalization (e.g., Batchnorm and Layernorm) are merged.
>
> When hyperparameter distributions differ largely between training and test samples, our LGPN leverages the GCN refinement block and the powerful GTC as the backbone to achieve a robust and generalizable model parsing ability.
> This is demonstrated by Table 1 of the main paper, in which we evaluate LGPN on unseen GMs and it shows SoTA model parsing performance. In addition, please refer to **Table R.6** from the response to **Reviewer nGEg**, in which more GMs are used for the evaluation.
>
> **Q4: Experiment section clarification:**
>
> The supplementary material offers details about the experimental setup and implementations. In the revised version, we will also polish the expression in the experiment section.

---

> > ### Comment · Reviewer_born · 2024-08-10
> >
> > Thank the authors for the response. All my questions have been answered clearly.

---

> > > ### Author Response · Authors · 2024-08-12
> > > **Replying to Review after rebuttal**
> > >
> > > Thank you so much for your positive feedback! It encourages us a lot!

---

### Official Review · Reviewer_nGEg · 2024-07-25

**Soundness:** 3
**Presentation:** 2
**Contribution:** 3
**Rating:** 5
**Confidence:** 3

**Summary:**

This paper solves the model parsing task that requires models to predicting hyperparameters of the generative models (GMs) through one input image. The proposed model LGPN predict hyperparameters and their dependencies via directed graphs. LGPN incorporates a learnable pooling-unpooling mechanism to convert this task as a node classification problem. As stated in paper, LGPN achieves state-of-the-art results in model parsing and some extended applications like coordinate attack detection.

**Strengths:**

1. As stated in the paper, LGPN consider more generative models in the training set, e.g., diffusion models. Therefore, LGPN can be adapted to the test environments where the images can be generated from diffusion model family.
2. Modeling model parsing as node classification on directed graphs makes sense. The proposed graph pooling network can group similar type of hyperparameters into supernodes, which indicates the model’s understanding on hyperparameter dependences.
3. LGPN show outstanding performance in experiments and has practical applications like coordinate attack detection.

**Weaknesses:**

1. The extensibility of LGPN can be limited in some aspects. For example, if a class of generative models that include new activation functions is proposed, LGPN may not be able to be directly applied to the new task environment.
2. Efficiency of the proposed method is an aspect that needs to be discussed. As the types of hyperparameters increase, the time cost of the algorithm may increase as well.

**Questions:**

1.	One concern is about the relation to prior works. The authors can better illustrate the contributions by discussing the relationship and difference between LGPN and the papers [1][2]. The paper [1] uses a meta model to predict model attributes from input data, and the paper [2] regards generative models as distributions of functions and captures model hyperparameters and architectures through the latent vector.
2.	How can we predict the continuous parameters of activation functions like leaky ReLU?
3.	Question about the problem setting. Will using more input images as context improves the accuracy in results?

[1] Oh S J, Schiele B, Fritz M. Towards reverse-engineering black-box neural networks[J]. Explainable AI: interpreting, explaining and visualizing deep learning, 2019: 121-144.

[2] Dupont, Emilien, Yee Whye Teh, and Arnaud Doucet. "Generative models as distributions of functions." arXiv preprint arXiv:2102.04776 (2021).

**Limitations:**

The ability of LGPN to infer input images generated with unseen or undefined model architectures is still unclear. For example, would LGPN predicts the leaky ReLU activation in the test environment if it is not appeared in the training data?

---

> ### Author Rebuttal · Authors · 2024-08-06
>
> We appreciate the reviewer's praise for our method's effectiveness and reasonable algorithm design. Here, we present answers to all concerns raised.
>
> **Q1: Difference between ours and two previous works:**
>
> We highlight three key differences between our approach and those in references [R1, R2], demonstrating that our proposed model parsing algorithm is more practical and better suited for general image forensic purposes.
>
> Three differences to [R1]:
>
> - [R1]'s algorithm is only evaluated on the MNIST dataset, while our evaluation dataset contains a wider variety of semantics, including hand-written digitals, faces, and generic objects.
>
> - [R1] only focuses on predicting architecture hyperparameters, yet we are capable of predicting both loss functions and architecture hyperparameters.
>
> - [R1]'s metamodel requires n query images to predict hyperparameters, while our LGPN requires only a single image as the input.
>
> Three differences to [R2]:
>
> - Unlike [R2], which uses estimated hyperparameters for generation purposes, we utilize these estimated hyperparameters for image forensic tasks.
>
> - While [R2] directly applies HyperNet [R4] for the hyperparameter prediction, we develope a more sophisticated LGPN with a high generalization ability.
>
> - [R2] focuses on estimating hyperparameters for a single architecture (the generator mentioned in [R2]), yet our approach is evaluated on 140 different Generative Models (GMs), many of which have distinctive architectures.
>
> **Q2: Multiple images input:**
>
> This is a valuable suggestion! Table R.5 reports results when the LGPN takes multiple images as inputs.
> Specifically, we denote input images and corresponding learned representations as {I1, I2 ... Ii} and {V1, V2 ... Vi}, respectively.
> We concatenate {V1, V2 ... Vi} for the final prediction.
> Table R.5 shows that more input images can possibly slightly improve LGPN's performance: inputting 12 images improves prediction on loss functions and continuous archi. hyperparameters but decreases in predicting discrete archi. hyperparameters.
> This means naively feeding LGPN with more inputs cannot provide substantial improvements, so a meaningful future research direction can be developing a specific module that effectively processes multiple images, such as meta-models as introduced in [R1] or ensemble learning methods.
> Secondly, when taking different numbers of images as inputs, our LGPN still maintains decent model parsing performance, indicating our approach's robustness.
>
> **Table R.5** Performance of using multiple input images.
>
> |Input Image Num.|Loss Fun.|Dis. Archi. Para.|Con. Archi. Para.|
> |:-:|:----------------:|:------------------:|:-----------------:|
> ||F1 / Acc(↑)|F1 / Acc(↑)|L1 error(↓)|
> |1|84.6/83.3|79.5/**77.5**|0.120|
> |4|84.1/83.2|**80.4**/74.3|0.118|
> |8|84.7/83.9|**80.4**/77.3|0.119|
> |12|**84.9**/**84.1**|79.2/77.1|**0.117**|
>
> **Q3: Generalization to the unseen generative model (GM) and undefined hyperparameters:**
>
> We agree with you on the importance of generalization ability to unseen GMs and hyperparameters! First, LGPN has effective extensibility and is capable of predicting 37 defined hyperparameters of unseen GMs.
>
> - Table 1 of the main paper reports performance on the RED140 dataset, where all models are evaluated on test sets that only have images from GMs unseen during the training.
>
> - Also, we use Table R.6 to report performances on 10 more unseen GMs (e.g., styleGANv3, controlNet1.1, etc.) that are exclusive to the RED140, showing that both FEN-PN and LGPN can deliver reasonable generalization performances.
>
> **Table R.6:** Model parsing performances on 10 more unseen GMs besides RED140.
>
> |Method|Loss Fun.|Dis. Archi. Para.|Con. Archi. Para.|
> |:-:|:----------------:|:------------------:|:-----------------:|
> ||F1/Acc(↑)|F1/Acc(↑)|L1 error(↓)|
> |FEN-PN|83.1/80.2|70.8/67.0|0.156|
> |Ours|85.0/84.1|75.0/71.5|0.131|
>
> Secondly, both FEN-PN [R3] and our work do not focus on predicting undefined hyperparameters beyond these 37 predictable hyperparameters.
> This is because the motivation of model parsing is to offer a new understanding to different GMs, thereby benefiting image forensic applications like coordinated attack detection. This understanding purpose is fulfilled by predicting 37 fundamental, predictable hyperparameters defined in [R3].
>
> Of course, predicting undefined hyperparameters, e.g., LeakyReLU, is an important future direction for the model parsing research topic. For example, based on our LGPN, one can add a few new graph nodes for undefined hyperparameters while keeping the original graph nodes—the learned dependency between graph nodes representing ReLU and LeakyReLU should be high.
>
> Lastly, we conduct an experiment (Table R.6.2) by naïve applying the pre-trained FEN-PN and LGPN to (i) classify if the input image is generated by GM with LeakyReLU and (ii) regress the continuous value about the negative slope used in LeakyReLU if available. The poor performance suggests that a trivial extension of existing model parsing methods is insufficient for undefined hyperparameter prediction, and a substantial new research is needed.
>
> **Table R.6.2:** Performances on predicting LeakyReLU.
>
> |Model|ACC(↑)| L1 error(↓)|
> |:------:|:---:|:---:|
> |FEN-PN|0.425|0.733|
> |LGPN|0.480|0.678|
>
> **Q4: Efficiency discussion:**
>
> Although the graph size might increase when adapting to new hyperparameters, our innovative graph pooling design actually largely reduces the graph size via the learned graph pooling mechanism, offering a more computationally efficient solution than prior GCN-based methods [10,7,59]. We implement LGPN with different input graph sizes, which are proportional to the number of hyperparameters. Then, we report the inference speed of one forward propagation as follows, indicating that the time cost is not largely affected by the increasing number of hyperparameters involved.
>
> |Graph Size|Speed|
> |----------|----------|
> |18|41.2 ms|
> |55|41.5 ms|
> |250|42.1 ms|
> |600|42.5 ms|

---

> > ### Author Response · Authors · 2024-08-12
> > **Thanks to Reviewer nGEg**
> >
> > Dear Reviewer nGEg:
> >
> > Please allow us to sincerely thank you again for reviewing our paper and providing valuable feedback, particularly for recognizing our algorithm's reasonable design, effectiveness, and high generalization ability as strengths.
> >
> > Please let us know if our response and additional experiments have properly addressed your concerns. We are more than happy to answer any additional questions during the discussion period. Your feedback will be greatly appreciated!
> >
> > Best,
> >
> > Paper14633 Authors

---

> ### Comment · Reviewer_nGEg · 2024-08-12
>
> Thank the authors for the response. The response have solved all my questions. I would like to maintain my score.

---

### Author Rebuttal · Authors · 2024-08-06

We want to thank all four reviewers for their valuable comments. We are delighted to see (a) **all reviewers** recognize our proposed LGPN's effectiveness and high generalization ability, (b) our research topic of model parsing is interesting and novel (**Reviewer born** and **Q98w**), (c) praises for our well-motivated and reasonable solution (**Reviewer nGEg** and **6bSd**), (d) the insightful ablation study has been appreciated by (**Reviewer 6bSd**), and (e) the useful proposed dataset for future work (**Reviewer Q98w**).

Moreover, we receive three positive feedbacks, and **Reviewer Q98w** kindly indicates the negative feedback can be changed after concerns are adequately addressed. Therefore, in this rebuttal, we provide precise answers to all questions, which mainly include (a) the difference between our work and two previous works (**Reviewer nGEg**), (b) additional experimental results on RED116 dataset (**Reviewer Q98w**), (c) additional generalization and robustness analysis (**Reviwer nGEg, born, and Q98w**), and (d) more elaborations on our experimental setup (**Reviewer 6bSd and Q98w**).

## 1. Recap:

Before we start our rebuttal, we would like to do a brief recap of our work. We study the research topic, model parsing [R3], in which the algorithm takes one generated image as the input and predicts 37 hyperparameters used in the generative model (GM) that generates the input image.
These 37 hyperparameters are defined in [R3], including discrete architecture hyperparameter, continuous architecture hyperparameter, and loss functions.
Motivated by the previous work's limitation in learning hyperparameter dependencies, we proposed a learnable graph pooling network (LGPN) that uses the GCN refinement block with a learnable pooling-unpooling mechanism to model hyperparameter dependencies, and a Generation Trace Capturing Network (GTC) effectively learns the generation trace from given generated images.
Our work achieves the SoTA performance on model parsing and has practical applications like coordinate attack detection.

## 2. Additional model parsing performance on RED116 dataset:

As suggested by **Reviewer Q98w**, we use Table R.1, R.2, and R.3 to show detailed performance comparisons to FEN-PN on the RED116 dataset for predicting continuous architecture hyperparameters, discrete architecture hyperparameters, and loss functions, respectively.
Also, Table R.4 illustrates the performance of face and non-face GMs.

These four tables, along with the main paper's Table 1, show that our LGPN surpasses FEN-PN [R3] on all metrics on both RED116 and RED140 datasets, indicating LGPN's better generalization ability and robustness in the model parsing task.

**Table R.1** Prediction performance on continuous architecture hyperparameters of RED116.

|Method\Metrics|L1 error(↓)|P-value(↓)|Corr. coef(↑)| Coef. of det.(↑)|Slope(↑)|
|:--------------------:|:---------:|:--------:|:-----------:|:---------------:|:-------:|
|**FEN-PN**|0.149±0.019|0.022±0.007|0.744±0.098 |0.612±0.161|0.921±0.021|
|**Ours**|**0.130**±**0.011**|N/A|**0.833**±**0.098**|**0.732**±**0.177**|**0.983**±**0.012**|


**Table R.2** Prediction performance on discrete architecture hyperparameters of RED116.

|Method\Metrics|F1(↑)|Acc(↑)|
|:--------:|:----:|:----:|
|**FEN-PN**|0.718±0.036|0.706±0.040|
|**Ours**|**0.743**±**0.033**|**0.755**±**0.030**|

**Table R.3** Prediction performance on loss functions of RED116.

|Method\Loss function|F1(↑)|Acc(↑)|
|:--------:|:----:|:----:|
|**FEN-PN**|0.813±0.019|0792±0.021|
|**Ours**|**0.841**±**0.011**|**0.833**±**0.017**|

**Table R.4** Performance of face GMs and non-face GMs of RED116.

|Method|Test GM|Train GM|Con. Archi. Para.|Dis. Archi. Para.|Loss function|
|:------:|---------|---------|:-----------------------------:|:-----------------------:|:-------------------:|
||||L1 error(↓)|F1(↑)|F1(↑)|
|FEN-PN|Face|Face|0.139±0.042|0.729±0.106|0.788±0.146|
|Ours|Face|Face|**0.112**±**0.028**|**0.786**±**0.116**|**0.801**±**0.134**|
|------|------|-------|--------------------|-------------------|-------------|
|FEN-PN|Face|Non-Face|0.213±0.066|0.688±0.125|0.759±0.1|
|Ours|Face|Non-Face|**0.139**±**0.063**|**0.694**±**0.117**|**0.771**±**0.2**|
|------|------|-------|--------------------|-------------------|-------------|
|FEN-PN|Face|Full|0.118±0.046|0.712±0.129|0.833±0.136|
|Ours|Face|Full|**0.099**±**0.044**|**0.745**±**0.099**|**0.840**±**0.123**|
|------|------|-------|--------------------|-------------------|-------------|
|FEN-PN|Non-Face|Face|0.118±0.021|0.794±0.110|0.864±0.094|
|Ours|Non-Face|Face|**0.116**±**0.016**|**0.810**±**0.102**|**0.870**±**0.092**|
|------|------|-------|--------------------|-------------------|-------------|
|FEN-PN|Non-Face|Non-Face|0.125±0.031|0.667±0.099|0.858±0.115|
|Ours|Non-Face|Non-Face|**0.100**±**0.027**|**0.692**±**0.101**|**0.882**±**0.112**|
|------|------|-------|--------------------|-------------------|-------------|
|FEN-PN|Non-Face|Full|0.082±0.045|0.832±0.046|0.886±0.061|
|Ours|Non-Face|Full|**0.080**±**0.042**|**0.844**±**0.032**|**0.901**±**0.021**|

## 3. Contents will be included in the revised version:

The revised version draft will include contents of the rebuttal to better reflect constructive suggestions received. Please refer to  **Q2** from the response to the **reviewer 6bSd**.

## 4. References:

**[R1]** S. J. Oh, B. Schiele, and M. Fritz, "Towards reverse-engineering black-box neural networks," in Explainable AI: Interpreting, Explaining and Visualizing Deep Learning, 2019.

**[R2]** E. Dupont, Y. W. Teh, and A. Doucet, "Generative models as distributions of functions," arXiv preprint arXiv:2102.04776, 2021.

**[R3]** V. Asnani, X. Yin, T. Hassner, and X. Liu, "Reverse Engineering of Generative Models: Inferring Model Hyperparameters from Generated Images," PAMI, 2023.

**[R4]** D. Ha, A. Dai, and Q. V. Le, "HyperNetworks," ICLR, 2017.

---

### Decision · Program_Chairs · 2024-09-25

**Decision:**

Accept (poster)

**Comment:**

This paper considers the model parsing task that predicts hyperparameters of the generative models through an input image. Compared to the existing work, this paper considers more generative models in the training set (e.g., diffusion models) and their dependencies via directed graphs. The reviewers recognize the strengths in the following aspects: 1) the considered problem is interesting and novel (Reviewers born, 6bSd, Q98w); 2) the approach is reasonable and practical (Reviewers nGEg, born, 6bSd); and 3) the proposed method is effective in terms of comparison performance (Reviewers nGEg, born, 6bSd, Q98w).

The authors have also addressed a number of concerns raised by the reviewers in the rebuttal and during the discussion period, such as the relationship and difference between the proposed method and the related works, the time cost as the types of hyperparameters increase, how the proposed method aids in the detection of coordinated attacks, the distinction between the two datasets, the sensitivities of hyperparameters, etc. These clarifications and supplementary experimental results should be reflected in the revised version.

All reviewers lean to accept this paper. However, some concerns not fully addressed (e.g., LGPN may not be able to be directly applied to the new task environment raised by Reviewer nGEg) can be further explored to improve the generalization ability.